# Analysis of $CO_2$ spatiotemporal variations in China using a weather-biosphere-online-coupled model

Xinyi Dong[1,2], Man Yue[1,2], Yujun Jiang[3,4], Xiao-Ming Hu[5], Qianli Ma[4], Jingjiao Pu[3], and Guangqiang Zhou[6]

[1]School of Atmospheric Science, Nanjing University, Nanjing, 210023, China
[2]Joint International Research Laboratory of Atmospheric and Earth System Sciences & Institute for Climate and Global Change Research, Nanjing University, Nanjing, 210023, China
[3]Zhejiang Meteorological Science Institute, Hangzhou 310008, China
[4]Zhejiang Lin'an Atmospheric Background National Observation and Research Station, Hangzhou 311307, China
[5]Center for Analysis and Prediction of Storms, University of Oklahoma, Norman, Oklahoma, 73072, USA
[6]Shanghai Key Laboratory of Health and Meteorology, Shanghai Meteorological Service, Shanghai, 200135, China

*Correspondence to:* Yujun Jiang (yjjiang@pku.org.cn) and Xiao-Ming Hu (xhu@ou.edu)

**Abstract.** Dynamics of atmospheric $CO_2$ has received considerable attention in the literature, yet significant uncertainties remain within the estimates of contribution from terrestrial flux and the influence of atmospheric mixing. In this study we apply the WRF-Chem model configured with the Vegetation Photosynthesis and Respiration Model (VPRM) option for biomass fluxes in China to characterize the dynamics of $CO_2$ in the atmosphere. The online coupled WRF-Chem is able to simulate biosphere processes (photosynthetic uptake and ecosystem respiration) and meteorology in one coordinate system. We apply WRF-Chem for a multi-year simulation (2016-2018) with integrated data from a satellite product, flask samplings, and tower measurements to diagnose the spatiotemporal variations of $CO_2$ fluxes and concentrations in China. We find that the spatial distribution of $CO_2$ was dominated by anthropogenic emissions, while its seasonality (with maxima in April 15 ppmv higher than minima in August) was dominated by terrestrial flux and background $CO_2$. Observations and simulations revealed a consistent increasing trend in column-averaged $CO_2$ ($XCO_2$) of 2.46 ppmv (0.6%/yr) resulting from anthropogenic emission growth and biosphere uptake. WRF-Chem successfully reproduced ground-based measurements of surface $CO_2$ concentration with mean bias of -0.79 ppmv and satellite derived $XCO_2$ with mean bias of 0.76 ppmv. The model-simulated seasonality was also consistent with observations, with correlation coefficients of 0.90 and 0.89 for ground-based measurements and satellite data, respectively. Tower observations from a background site Lin'an (30.30°N, 119.75°E) revealed a strong correlation (-0.98) between vertical $CO_2$ and temperature gradients, suggesting a significant influence of boundary layer thermal structure on the accumulation and depletion of atmospheric $CO_2$.

## 1 Introduction

Climate research requires accurate characterization of atmospheric $CO_2$, which is closely affected by the both atmospheric transport and terrestrial sources and sinks (Bauska et al., 2015; Keenan et al., 2016). Our current knowledge largely comes from interpreting ground- or space-based measurements and model simulations. While observation is limited by spatial and temporal coverages, modelling approaches also suffer from various uncertainties (Shi et al., 2018). Modelling assessment of $CO_2$ is usually conducted through two methods: first, process- or data-driven biosphere models in which terrestrial fluxes are diagnostically calculated with theoretical functions (Tian et al., 2015) or determined through semi-empirical relationships derived from ground measurements and/or satellite products with machine learning techniques (Papale and Valentini, 2003); second, inverse modelling in which prior flux estimates applied in atmospheric transport models are adjusted by observational data and/or satellite products to determine posterior flux (Peylin et al., 2002;Kountouris et al., 2018). Process-driven biosphere models have difficulties capturing spatial and temporal variabilities at fine resolution because parameters calibrated from a limited number of site observations are applied across a variety of land covers (Todd-Brown et al., 2013). Atmospheric inverse modelling is predominantly affected by the presumed prior flux, and different assimilation techniques can give different and even conflicting results (Peylin et al., 2013). These fundamental features highlight the limits of these approaches for accurately modelling carbon dynamics.

Researchers have attempted to reconcile differences between "bottom-up" biosphere models and "top-down" atmospheric inverse models, and recent studies have demonstrated increasing levels of agreement owing to improved understanding  of both approaches, such as better parameterization of biosphere processes (Dayalu et al., 2018), more accurately constrained estimates of prior flux (Crowell et al., 2018;Feng et al., 2019), and advanced measurement/satellite instruments that provide high quality data for assimilation (Gaubert et al., 2019); however, critical model disagreements still persist (Kondo et al., 2020). To bridge the gap between terrestrial flux and atmospheric mixing, a type of weather-biosphere coupled model (Ahmadov et al., 2007;Mahadevan et al., 2008) has been developed to simulate biosphere processes and meteorology conditions in one coordinate system, allowing their interactions to be properly addressed. Previous modelling studies (Ahmadov et al., 2009;Kretschmer et al., 2012;Park et al., 2018;Beck et al., 2013;Park et al., 2020;Pillai et al., 2012) have demonstrated the weather-biosphere coupled model can successfully capture the mesoscale $CO_2$ transport at regional and local scales with significant improvements. But whether it can reproduce the long-term variations and subsequently estimate carbon fluxes at regional scales with high confidence remains a crucial issue to be addressed.

Understanding the spatiotemporal characteristics of atmospheric $CO_2$ is a key priority in China because of the central role it plays in regulating the climate and environment. In recent years, tremendous efforts have been made in China to control anthropogenic emissions from fossil fuel combustion for both air quality and climate mitigation purposes (Zheng et al., 2018). While the sources and sinks of air pollutants have been thoroughly examined and well documented (Huang et al., 2020), the dynamics of $CO_2$ at regional to national scales remain poorly understood due to lack of long-term observations and limited modelling studies (Han et al., 2020). Li et al. (2020) applied a weather-biosphere model with tower observations to analyse

CO$_2$ fluxes and concentrations over mixed forest and rice paddy in northeast China, but the one-year simulation limits the attempt to investigate interannual CO$_2$ variation which is subject to substantial change (Fu et al., 2019b). Wang et al. (2019) applied satellite products and in-situ observations with inverse modelling to derive posterior carbon fluxes and reported 100% uncertainty for constraining global terrestrial flux. Fu et al. (2020) applied GEOS-Chem simulation with offline Carbon Tracker (Peters et al., 2007) as input to estimate impacts of terrestrial flux and anthropogenic emissions on the annual variation

of CO$_2$ concentrations, but regional-scale assessment was limited by coarse grid resolution (2°×2.5°). Machine-learning technique has also been employed to upscale site observations to regional-scale (Yao et al., 2018; Zhu et al., 2014), but the estimations of carbon budget and dynamics retain large uncertainty due to the diversity of biomass among sites and coarse grid resolution. These pilot studies have shed light on improving the understanding of spatiotemporal characteristics of CO$_2$ in China with modelling or observational methods, but an integrated investigation with both modelling and observations at fine-

scale is urgently needed to expand diagnostic understanding of localized and regional transport, flux, and concentration of CO$_2$ to inform emission management and climate adaption policies (Fu et al., 2019a;Niu et al., 2017;Wang et al., 2019).

In this study we use the WRF-Chem model configured with the Vegetation Photosynthesis and Respiration Model (WRF-Chem) option (Hu et al., 2020;Mahadevan et al., 2008) to simulate and characterize the spatiotemporal variation of atmospheric CO$_2$ in China from 2016-2018, and also to validate this weather-biosphere model with recent advanced satellite and tower

observations. WRF-Chem has been applied in a few case studies over the United States (Hu et al., 2020), Europe (Kretschmer et al., 2012), northeast China (Li et al., 2020), and South Korea (Park et al., 2020); this study attempt to apply and evaluate it for a multi-year simulation over China. We first describe the modelling methods and data followed by model validation against observations from multiple datasets, and then present the spatiotemporal variations and estimates of contributions from anthropogenic emissions, terrestrial flux, and background concentrations. Finally, we investigate tower data and reveal the

boundary layer thermal structure impacts on atmospheric CO$_2$ accumulation and depletion.

## 2 Method

We conduct nested WRF-Chem (Version 3.9.1.1) simulations over China (domain shown in Fig.1(a)) and Yangtze River Delta (YRD) region (domain shown in Fig.1(d)) at 20 km and 4 km grid resolution, respectively. Both simulations were configured with 47 vertical layers with model tops at 10hPa. Model configuration in this study followed the work by Hu et al. (2020) and

90 Li et al. (2020). We applied the YSU planetary boundary layer (PBL) scheme (Hong et al., 2006), Morrison microphysics (Morrison et al., 2009), Duhia short-wave radiation (Dudhia, 1989), RRTM long-wave radiation (Mlawer et al., 1997), Grell-3 cumulus scheme (Grell and Devenyi, 2002), and Noah land-surface scheme (Chen and Dudhia, 2001), with more details summarized in Table S1. In general, the 4km-grid simulation showed no significant difference as compared to the 20km-grid simulation (demonstrated in Figure S1 and Figure S2), thus the 20km-grid simulation was used to characterize the

95 spatiotemporal distributions of CO$_2$ over China, and the 4km-grid simulation was only used to compare with tower data collected at a background site in YRD. Discussions in the next section will mostly refer to the 20km-grid simulation unless

otherwise specified. Initial and lateral boundary conditions for the 20km-grid simulations were derived from the mole fraction product of CarbonTracker (Peters et al., 2007) with $3° \times 2°$ resolution. The latest update of column average $CO_2$ ($XCO_2$) concentration assimilation product from CarbonTracker (CT2019) with $1° \times 1°$ resolution (Jacobson, 2020) was also employed to compare with the WRF-Chem simulation. The anthropogenic emission inventory is from the Open-source Data Inventory for Anthropogenic $CO_2$ (ODIAC) with $0.1° \times 0.1°$ resolution (Oda et al., 2018) shown in Fig.1(a). ODIAC has been widely applied in recent modelling studies and demonstrated good agreement with other global inventories (Hedelius et al., 2017;Hu et al., 2020). Ocean flux is from climatology estimation (Takahashi et al., 2009); and vegetation fractions and enhanced vegetation index (EVI, shown in Fig.1(b)) are from MODIS (Huete et al., 2002). $CO_2$ from initial and boundary conditions, anthropogenic emission, and terrestrial biogenic flux were tagged as BCG, ANT, and BIO, respectively, to allow the contributions from each process to be identified and quantified through one simulation.

WRF-Chem calculates ecosystem respiration (ER) and gross ecosystem exchange (GEE) with the following functions as:

$$ER = \alpha \times T + \beta \qquad (1)$$
$$GEE = -\lambda \times T_{scale} \times W_{scale} \times P_{scale} \times (1 + PAR/PAR_0)^{-1} \times EVI \times PAR \qquad (2)$$

where T is the air temperature at 2m above the surface (T2); $\alpha, \beta, \lambda$ are vegetation type-dependent parameters; $PAR_0$ is the vegetation type-dependent half-saturation value of photosynthetically active radiation (PAR); and $T_{scale}, W_{scale}, P_{scale}$ are scaling factors for temperature, water stress, and phenology, respectively. In this study we take the atmosphere as a reference, thus GEE has a negative sign and ER has a positive sign. The current version of WRF-Chem is parameterized ($\alpha, \beta, \lambda$) for 7 vegetation types (Fig.1(c)): crops, mixed forest, evergreen forest, deciduous forest, shrub, savanna, and grass. For each modelling grid, ER and GEE are calculated as the weighted averages of each vegetation type based on their fractional abundance. Recent studies (Hu et al., 2020;Li et al., 2020) have investigated the uncertainty associated with this parameterization through sensitivity simulations and suggested the crops can be further divided into subcategories based on eddy-covariance (EC) tower measurement to improve the model. In this study we used the default parameterization (values presented in Table S2), which has been demonstrated to successfully reproduce the terrestrial flux over northeast China (Li et al., 2020). In contrast, CT2019 applies a process based biosphere model, the Carnegie-Ames Stanford Approach (CASA(Zhou et al., 2020)), driven by year-specific weather and satellite data to simulate terrestrial fluxes (Peters et al., 2007). CASA also estimates photosynthetic uptake based on solar radiation and plant phenology, and estimates respiration as a function of T2. CASA directly simulates monthly means of Net Primary Production (NPP) and heterotrophic respiration ($R_H$). NPP is the difference between photosynthetic uptake (equivalent to GEE) and autotrophic respiration ($R_A$). The summary of $R_H$ and $R_A$ is equivalent to ER. Thus, WRF-Chem and CASA are essentially very similar in terms of considering methodology impact; however, it should be noted that to resolve CASA simulated NPP into GEE and $R_A$, CT2019 applies the assumption that GEE is twice that of NPP, which implies that for the same plants the photosynthetic carbon uptake is double the magnitude of autotrophic respiration (but of opposite sign). This assumption is applicable at monthly scale but may have difficulty to

reproduce the rapid changes at hourly and daily scales due to impact from weather systems, which will be demonstrated with
more details in Section 3.2.

Hourly measurements of $CO_2$ concentrations were collected at the Lin'an Regional Atmospheric Background Station (30.30˚N, 119.75˚E, surroundings shown in Fig.2(a)) with Picarro G1301 and G1302 trace gas analysers mounted on an observation tower at 21 and 55 meters, respectively, above ground level (AGL) and analysed online (data analysis lab shown in Fig.2(b)). The station is located in the remote area of Hangzhou 138.6 meters above sea level in the middle of a hilly area covered by
mixed forest. The observation tower is 60km to the west of downtown center of Hangzhou and 195km to the southwest of Shanghai. Fig.2(c) and (d) presents the wind rose map at Lin'an derived from hourly observations of 10m and 55m wind respectively, which clearly shows the northeast and southwest as prevailing wind directions. The station can properly represent the background atmospheric environment in YRD as demonstrated in previous studies (Deng et al., 2018;Pu et al., 2020). The tower data provides a representative sampling of $CO_2$ gradients resulting from exchange between atmosphere mixing and
terrestrial flux.

Atmospheric samples near the surface were collected at monthly intervals and analysed for $CO_2$ through the National Oceanic and Atmospheric Administration's (NOAA's) Earth System Research Laboratory (ESRL) at four sites (locations shown in Fig.1(a)) within our study domain, including Dongsha Island (DSI, 20.69˚N, 116.73˚E), Lulin (LLN, 23.47˚N, 120.87˚E), Ulaan Uul (UUM, 44.45˚N, 111.09˚E), and Mt. Waliguan (WLG, 36.29˚N, 100.89˚E). The Orbiting Carbon Observatory-2
(OCO-2) satellite product (Kiel et al., 2019) with daily intervals was employed to validate simulation of column averaged $CO_2$ ($XCO_2$) concentrations. A total of 204,940 OCO-2 version9 swath data covering the simulation period was used in this study. Daily ground-based Fourier transform spectrometer (FTS) Measured $XCO_2$ at Hefei site (31.90˚N, 117.17˚E) was also collected through the Total Carbon Column Observing Network (TCCON) for year 2016 (Wang et al., 2017). The TCCON-Hefei site was located in the northwestern rural area of Hefei city and measurements were conducted from September 2015 to
December 2016 (Liu, 2018). WRF has been evaluated extensively and consistently performs well for reproducing the meteorology fields and the transport of atmospheric tracers in China (Gao et al., 2015;Tang et al., 2016;Wang et al., 2017;Yang et al., 2019), so this study will only present the simulation performance for $CO_2$ which hasn't been thoroughly discussed in the literature.

## 3 Result and Discussion

## 3.1 Model evaluation

We first evaluate the capability of WRF-Chem to reproduce concentrations of surface $CO_2$ and $XCO_2$, and we find fairly good model performance through the comparison with satellite and ground-based observations. The WRF-Chem simulated surface layer (mid-level height AGL is 12m) $CO_2$ and $XCO_2$ averages between 2016-2018 are demonstrated in Fig.3(a) and (b) respectively. High concentrations were found over industrial areas such as the North China Plain (NCP), Pearl River Delta

(PRD), and Yangtze River Delta (YRD), where the surface $CO_2$ and $XCO_2$ were above 440 ppmv and 408 ppmv, respectively; the domain averages were 411 ppmv and 406 ppmv, respectively. While most climate models assume evenly distributed $CO_2$ (Fung et al., 1983;Kiehl and Ramanathan, 1983), our data demonstrates a prominent gradient between industrial and remote areas (e.g., Tibet Plateau, Mongolia), especially for surface $CO_2$, which could be an important source of uncertainty for estimating the long-wave radiation effect (Xie et al., 2018). Spatial patterns of $CO_2$ and $XCO_2$ were in close agreement with

ODIAC, indicating the dominant impact of anthropogenic emission in determining the $CO_2$ distribution. WRF-Chem simulated $CO_2$ was generally consistent with CT2019 (Fig.3(c)), but CT2019 estimated near surface $CO_2$ (mid-level height AGL is 25m) over the coastal industrial areas YRD and PRD because the ocean module used in CT2019 estimated stronger air-sea exchange than the ocean flux by Takahashi et al. (2009) used in WRF-Chem. The two models showed better agreement for $XCO_2$ (Fig.3 (b) and (e)), but also differed by ~1 ppmv over Taklamakan Desert and along the eastern side of the Tibet Plateau. The OCO-

2 swath data were integrated into the corresponding horizontal grids of WRF-Chem and CT2019 respectively, to validate $XCO_2$. Biases of WRF-Chem and CT2019 both fall into the range of $\pm3$ ppmv as shown in Fig.3(c) and (f), respectively, but WRF-Chem apparently provided more details of spatial gradient. WRF-Chem showed well-mixed underestimations and overestimations along neighbouring satellite tracks, while CT2019 tended to overestimate (underestimate) over Tibet Plateau (Taklamakan Desert) where WRF-Chem gave slightly smaller biases. Fig.4(a) and (b) present the raw data pairs between

models and OCO-2 with daily interval for WRF-Chem and CT2019, respectively. In general, the WRF-Chem model reproduced OCO-2 well, with mean bias (MB) of 0.76 ppmv, and CT2019 showed MB of 0.54 ppmv, suggesting an overall acceptable performance of the weather-biosphere model to simulate the spatial distribution pattern of $XCO_2$ in China.

We further analyse WRF-Chem validation against OCO-2 for the seven vegetation types in each season and find no prominent difference (evaluation statistics summarized in Table 1). Regarding vegetation type, the model showed the largest MB over

deciduous forest of -1.01 and 1.27 ppmv in summer and winter, respectively, which only covered a very small portion in northeast China. The three most abundant coverage vegetation types in China are grass, crops, and mixed forest. $XCO_2$ simulated by WRF-Chem over grass areas was slightly overestimated by 0.31~0.68 ppmv throughout the year, and the MB over mixed forest was -0.43~0.59 ppmv, indicating a good performance of the model over the vast majority of areas of China. Performance over crops generally showed larger discrepancy than other vegetation types, with MB ranging from 0.66 ppmv

in summer to 1.19 ppmv in winter, suggesting the model tends to slightly overestimate column concentration of $CO_2$ over cropland. Li et al. (2020) reported that WRF-Chem underestimated biosphere carbon over rice paddy sites (by ~3%) in northeast China and suggested the parameterization of $\alpha, \beta, \lambda$ as the most important cause. Cropland differs significantly across China with various types of species such as rice, wheat, and corn, for which literatures reported substantially different rates of ecosystem respiration and photolysis uptake (Gao et al., 2018;Yang et al., 2016;Zhu et al., 2020). Thus, applying one set of

parameters to represent all crops may be responsible for the lingering uncertainty of simulated $XCO_2$. In terms of seasonal difference, WRF-VRPM showed slightly smaller bias in summer and larger bias in winter, and the correlation coefficients

were all ~0.8, consistent with application over the U.S. (Hu et al., 2020) which also reported slightly better performance in summer than other seasons.

Fig.4 also presents the overall simulation bias against ground-based observations at their raw temporal intervals (monthly for data at ESRL sites, hourly for tower data at Lin'an, and daily for TCCON at Hefei). At the ESRL sites (Fig.4(c)), surface $CO_2$ concentrations were simulated well with minor overestimation by 0.69 ppmv. Evaluation at the Lin'an station was performed with the 4km-grid simulation. The mid-level heights of WRF-Chem's first, second, and third layers were 12.3m, 36.9m, and 61.6m, respectively, and simulations were linearly interpolated to 21m and 55m to compare with the tower data. The evaluation at 21m AGL (Fig.4(d)) shows slight overestimation by 0.02 ppmv, but the evaluation at 55m height (Fig.4(e)) shows relatively large overestimations by 1.06 ppmv. The discrepancy is likely due to the combined effect of vertical allocation of anthropogenic emission (Brunner et al., 2019) and parameterization of VPRM. Tracer transport models (such as WRF-Chem and CASA) and inverse modelling methods allocate anthropogenic $CO_2$ emission into the near surface layer due to lack of injection height information, which may subsequently lead to systematic overestimation of surface $CO_2$ concentration in industrial areas. Through a regional scale (750×650km) modelling study around the city of Berlin, Brunner et al. (2019) reported that distributing anthropogenic emission into the surface layer overestimated near-surface $CO_2$ concentration by 14% in summer and 43% in winter as compared with considering the vertical profiles of local anthropogenic sources. Lin'an observation tower is located at a densely vegetated area. Validation against OCO-2 suggested that WRF-Chem did not show significantly different performance over different vegetation types as shown in Table 1. As compared to the ESRL background sites which were located in more remote areas with little anthropogenic emission (Fig.1(a)), Lin'an was more frequently affected by regional anthropogenic emissions which were transported from Shanghai and Hangzhou due to the prevailing northeast wind (Pu et al., 2014), indicating that the emission allocation discrepancy may induce more prominent error at Lin'an. In fact, the 20km-grid WRF-Chem simulation bias at Lin'an were 5.34 and 5.41 ppmv at 21m and at 55m respectively (Figure S2), significantly larger than the bias at ESRL sites. In addition, both the 20km-grid and 4km-grid simulations showed relatively larger bias at 55m than 21m due to smaller topography roughness and higher wind speed which increases with height according to observations (Figure S3). CT2019 also substantially overestimated at Lin'an, but the first, second, and third layers' mid-level heights are 25m, 103m, and 247m, respectively, so we do not present the direct comparison with the tower data. Simulated $XCO_2$ from both WRF-Chem and CT2019 were well consistent with the TCCON Hefei site observations as shown in Fig.4(f), with MB by -0.79 ppmv and -0.78 ppmv respectively, and NMB by -0.20% and -0.19% respectively. The 4km-grid simulation showed very similar result to the 20-grid simulation for $XCO_2$ (Figure S1 and Figure S2). Recent atmospheric inverse modelling studies (Fu et al., 2019a;Wang et al., 2019;Xie et al., 2018) reported the simulation bias of $XCO_2$ as 0.5-2 ppmv with posterior flux inputs. The WRF-Chem model applied in this study has demonstrated good agreement with the observations though our evaluation.

## 3.2 CO₂ seasonal variation and trend in China

We next analyse the seasonality of $CO_2$ and $XCO_2$ and find that the terrestrial flux played a more influential role than anthropogenic emission. WRF-Chem successfully reproduced seasonal variations of $CO_2$ at ESRL sites, with a correlation coefficient of 0.90 (Fig.5(a)). The WRF-Chem 4km-grid simulation showed a correlation coefficient of 0.82 with the Lin'an tower observation (averaged for daytime 21m and 55m data). Both the model and measurements showed prominent seasonal cycles for surface $CO_2$ concentrations. The WRF-Chem simulation showed maxima in April (413-419 ppmv) and minima in August (399-404 ppmv) as presented in Fig.5(b). The model suggested that the anthropogenic $CO_2$ contribution was 2.6 ppmv in both months, while the biogenic contributions were 3.1 and -1.2 ppmv in April and August, respectively (Fig.5(d)). Anthropogenic emission (Fig.5(f)) showed a flat curve with relatively higher values in December due to fuel combustion for heating (Zheng et al., 2018). EVI showed maxima in July and August (Fig.5(f)). During summer, photosynthetic uptake almost completely compensated anthropogenic emission, causing the minimum $CO_2$ concentration observed in August, while the higher anthropogenic emission in December and respiration flux in April led to the two corresponding peaks. The anthropogenic $XCO_2$ contributions were 0.5 and 0.6 ppmv in April and August, respectively, and the biogenic contributions were 0.8 and -1.5 ppmv, respectively, suggesting that the seasonality of $XCO_2$ was also primarily dominated by terrestrial flux. Furthermore, the seasonality at high-latitude ESRL sites (UUM and WLG) was stronger than at Lin'an and low-latitude sites (DSI and LLN) because of the larger temperature and photosynthetically active radiation (PAR) gradients. Annual average anthropogenic and biogenic $XCO_2$ contributions were 7.1 and -1.9 ppmv, respectively, indicating that biosphere uptake was an important carbon sink offsetting 27% of anthropogenic emission and slowing the growth of atmospheric $CO_2$.

$XCO_2$ showed similar seasonality, with minima in August and maxima in April and December (Fig.5(b)). Both WRF-Chem and CT2019 showed good agreement with TCCON Hefei observations with correlations of 0.89 and 0.88, respectively (Fig.5(e)). However, we note that WRF-Chem simulated drastic changes (e.g., the grey shaded period in Fig.5(e)) that were not reproduced by CT2019. Fig.6 shows the daily concentrations of $XCO_2$ overlaid with horizontal wind speed at 10m AGL from WRF-Chem and CT2019 and highlights large discrepancies over Hefei (Figure S4 shows the same comparison but using WRF-Chem 4km-grid simulation data). Between April 1st and 3rd 2016, an 850 hPa trough associated with a surface cold front moved southeastward from Mongolia to the North China Plain (NCP) (weather maps shown in Fig.6(g)-(i)). At the leading edge of the front, a convergence zone associated with a low pressure center formed, which led to significant cloud formation and subsequently reduced short-wave radiation. As a result, photosynthetic carbon uptake was reduced, leading to enhancement of atmospheric $CO_2$. Meanwhile, the cold front transported anthropogenic $CO_2$ from NCP to YRD, and the convergence zone along YRD ahead of the front facilitated the accumulation of air pollutants and $CO_2$ from anthropogenic emissions. With its coarse spatiotemporal resolution, CT2019 had difficulty reproducing such regional weather systems that can lead to rapid and localized changes in $CO_2$ concentration and terrestrial flux, indicating the importance of fine resolution modelling to better represent the small spatial scale and rapid temporal scale variations of $CO_2$ (Agusti-Panareda et al., 2019).

We also find a notable increasing trend for the 3-year study period. Observed $CO_2$ annual enhancement was 2.2 ppmv/yr (0.56%/yr) at the ESRL sites and 2.3 ppmv/yr (0.54%/yr) at Lin'an. The observed average $CO_2$ concentrations at Lin'an (428 ppmv) were substantially higher than those at ESRL sites (407-410 ppmv). The prominent higher levels of $CO_2$ and slightly higher absolute growth rate at Lin'an can be attributed to the influence of the transport regional anthropogenic emission which is growing at rate of 0.82%/yr as suggested by ODIAC. Domain-wide $XCO_2$ was also found to increase by 2.3 ppmv/yr (0.57%/yr) as suggested by OCO-2 and 2.5 ppmv/yr (0.61%/yr) as suggested by the simulation. WRF-Chem reproduced the trends in good agreement with ground and satellite observations. Model simulated budgets suggested that the increasing trends for anthropogenic, biogenic, and background $XCO_2$ were 0.81%/yr, -9.17%/yr, and 0.59%/yr, respectively; the trends for anthropogenic, biogenic, and background $CO_2$ were 4.95%/yr, -0.73%/yr, and 0.59%/yr, respectively. Our findings are consistent with recent measurements and inverse modelling studies but provide process-based estimates for anthropogenic emission and terrestrial flux. Wu et al. (Wu et al., 2012) reported measured $CO_2$ concentration at Changbai Mountain forest site in northeast China increased by 1.76 ppmv/yr between 2003-2010. With the atmospheric inversion modelling method, Fu et al. (2019b) estimated surface $CO_2$ in East Asia increased by 2-3 ppmv/yr between 2004-2012. These trends suggest that although anthropogenic emission increases at a steady rate in East Asia, photosynthetic uptake also serves as an increasing carbon sink due to enhanced EVI (0.29%/yr). However, as the interannual variability (IAV) of terrestrial flux is usually critically large and is affected by both vegetation itself and climate conditions (Fu et al., 2019b;Niu et al., 2017), simulation over longer time periods is necessary in future studies to conclusively comment on the changing trend of biosphere $CO_2$ in China.

### 3.3 Diurnal variation of near-surface $CO_2$ and influence factors

Finally, we examine the diurnal variation of $CO_2$ data at Lin'an station as shown in Fig.7 to reveal the temporal dynamics and atmospheric mixing of $CO_2$ at local scale. While both 21m (Fig.7(a)) and 55m (Fig.7(b)) $CO_2$ show prominent diurnal changes, the variations were larger in summer (JJA) than winter (DJF) and were larger at 21m than 55m, indicating the dominant influence of terrestrial flux over anthropogenic emission in determining the near surface $CO_2$ concentration. Fig.7(c) and (d) present the WRF-Chem simulation bias at 21m and 55m respectively, and Fig.7(e) and (f) present the bias of CT2019 at 21m and 55m respectively. We find that both models prominently overestimated during nighttime, which shall be attributed to the bias in simulating NEE. Li et al. (2020) reported the model overestimated nighttime NEE at a mixed forest site Wuying (47.15°N, 131.94°E) by 34% during the growing season (May-Sep.) according to eddy-covariance tower measurement. Fig.7(g) and (h) present the simulated NEE by WRF-Chem and CT2019, respectively, which show close correlations with the $CO_2$ simulation biases. While Lin'an is also covered by mixed forest, our evaluation suggests that WRF-Chem may also overestimate nighttime ecosystem respiration at Lin'an as it has a warmer climate condition than Wuying (Figure S5), and CT2019 has even greater bias for presenting the diurnal cycles of terrestrial flux.

We also find that planetary boundary layer height (PBLH) significantly affects diurnal accumulation and depletion of atmospheric $CO_2$ as shown in Fig.8(a). During daytime in the growing season, photosynthetic uptake results in lower $CO_2$ concentration; meanwhile, PBLH is also high and allows rapid vertical mixing between near surface and upper air. During nighttime when photosynthesis stops, $CO_2$ from ecosystem respiration starts to accumulate in the shallow stable boundary layer,

while the residual layer remains largely decoupled. Thus, atmospheric constituents with surface sources normally exhibit a vertical profile in which concentrations decrease with height in the stable boundary layer (Hu et al., 2020;Hu et al., 2012). Such boundary layer characteristics are confirmed by $CO_2$ vertical gradients at Lin'an in this study. $CO_2$ at 55m height was consistently lower than the near surface air at 21m during nighttime due to accumulation of respired $CO_2$ in the stable boundary layer. As photosynthetic uptake depleted the near surface $CO_2$ and daytime boundary layer convection developed, the $CO_2$

gradient was gradually weakened from 06:00 to 11:00 LT and remained minimal through the rest of the daytime; at midday when photosynthesis reaches maximum intensity, $CO_2$ at 21m was even lower than at 55m. WRF-Chem roughly reproduced the diurnal profile but noticeably underestimated the intensity of nighttime $CO_2$ difference ($\Delta CO_2$) likely due to the bias for simulating night time terrestrial flux as discussed above or underestimation of nighttime boundary layer stability by the PBL scheme (Hu et al., 2012).

The relationship between the near-surface $CO_2$ profile and boundary layer stability is further statistically examined. Fig.8(b) presents the correlation between air temperature gradient ($\Delta T/\Delta H$) and $CO_2$ concentration gradient ($\Delta CO_2/\Delta H$) calculated with diurnal profiles of tower observations averaged for 2016-2018, where $\Delta T$, $\Delta H$, and $\Delta CO_2$ is the differences of temperature, height, and $CO_2$ concentration between the two tower levels, respectively. Fig.8(b) clearly demonstrates the influence of boundary layer stability on the $CO_2$ vertical profile, as the correlation between $\Delta T/\Delta H$ and $\Delta CO_2/\Delta H$ reaches -0.98. On one

hand, a more stable PBL with a strongly positive temperature gradient would promote surface $CO_2$ accumulation and lead to a strongly negative $CO_2$ gradient, especially under inversion conditions when upper air has higher temperature (orange area in Fig.8(b)). Conversely, a strongly negative temperature gradient indicates stronger radiation, and subsequently greater photosynthesis and $CO_2$ depletion in the near surface layer, which would result in a positive $CO_2$ gradient (green area in Fig.8(b)) implying a lower $CO_2$ concentration at the surface. While the diurnal variations of $\Delta CO_2$ were primarily dictated by local

biogenic $CO_2$ fluxes and boundary layer dynamics, the two minor daytime peaks of $\Delta CO_2$ at Lin'an, at 10:00 and 18:00 LT (Fig.8(a)) likely suggest influence of transport of $CO_2$ from urban plumes in the region; for example, from Hangzhou which is 60 km away from the tower. Due to rush-hours anthropogenic emissions, $CO_2$ enhancement at Hangzhou relative to a background site exhibited a prominent bimodal curve with two peaks during early morning and early evening (Pu et al., 2018). Depending on meteorological conditions, particularly wind fields, urban $CO_2$ plumes from cities such as Hangzhou may be

transported to the Lin'an site. The influence of boundary layer conditions on $CO_2$ variability has been discussed in several studies through analysis of mountain site ground-based observations (Arrillaga et al., 2019;Esteki et al., 2017;Li et al., 2014), but our study applied tower data as direct evidence to demonstrate the significant impact of PBL thermal structure, which has rarely been documented. More importantly, although WRF-Chem failed to capture the bimodal $\Delta CO_2$ peaks at rush hours,

because monthly ODIAC data lacked an hourly profile, our analysis reveals the critical importance of careful configuration of the PBL scheme and spatiotemporal distribution of anthropogenic emission for weather-biosphere modelling of atmospheric $CO_2$.

## 4 Summary and Conclusions

In this study, the spatiotemporal variations of $CO_2$ in China are investigated with measurements from multiple datasets and a weather-biosphere coupled model simulation for 2016-2018. We find consistent higher concentrations over industrial areas with excessive anthropogenic emission and lower concentrations over densely vegetated areas. Observed $CO_2$ concentrations at Lin'an (427 ppmv) are significantly higher than remote ESRL sites (408 ppmv) although they are all established as "background" stations, indicating the dominant influence of anthropogenic emission at regional scales. The Lin'an tower data shows a large negative correlation (-0.98) between vertical $CO_2$ concentration and air temperature gradients, suggesting the significant influence of boundary layer stability on $CO_2$ accumulation and depletion. The online coupled weather-biosphere model WRF-Chem enables process-based estimations of contributions from anthropogenic emission (0.59 ppmv (0.15%)), terrestrial flux (0.16 ppmv (-0.04%)), and background concentration (405.70 ppmv (99.89%)) to average total $XCO_2$. Respective simulation biases of surface $CO_2$ and $XCO_2$ are 0.69 and 0.76 ppmv against ESRL site observations and OCO-2 satellite product with correlations of 0.87 and 0.90, indicating overall good performance of the WRF-Chem model. Maximum $CO_2$ concentrations are found in April and minima are found in August for all three years, and the seasonality is reproduced well by the model, which also reveals that terrestrial flux and background concentration dominated the seasonality rather than anthropogenic emission.

A steadily increasing trend in $XCO_2$ by 2.46 ppmv (~0.6%/yr) during the study period is demonstrated consistently by both model simulation and satellite product. Budget analysis suggests that anthropogenic emission increased by 0.83%/yr contributing to the 0.81%/yr growth rate of anthropogenic $XCO_2$ enhancement, 27% of which was offset by biosphere uptake. It is noted that terrestrial flux has significant inter-annual variability, thus a more robust estimation of the terrestrial flux trend should be obtained through a long-term study in the future. The background $XCO_2$, representing contributions from global circulation, increased by 2.37 ppm (0.59%/yr), suggesting that $CO_2$ level in China was growing at the same rate as the rest of the world.

The most significant modelling bias is identified from validation against the Lin'an tower 55m observations, which WRF-Chem 4km-grid simulation overestimated by about 1.06 ppmv with a correlation coefficient of 0.82. The allocation of anthropogenic emission into the surface layer is partially responsible for this modelling bias because Lin'an is closely affected by upwind industrial mega cities in YRD, suggesting the need to include vertical profiles of fossil fuel combustion to properly redistribute the ODIAC for modelling purposes. In addition, diurnal variations of the bias suggest that the modelling discrepancy is also induced by large uncertainty associated with simulating nighttime ecosystem respiration. Representation

and parameterization of photosynthetic carbon uptake in VPRM has been continuously improved during the past 10 years since its first release (Hu et al., 2020), but ecosystem respiration parameterization is still too simplified to fully represent the autotrophic and heterotrophic respiration of biomass(Hu et al., 2021). Li et al. (2020) and our study both reveal the urgent need to better calibrate VPRM parameterization over different vegetation types in China, and other methods such as inverse modelling is necessary to further validate the anthropogenic fluxes from ODIAC. Nevertheless, WRF-Chem is demonstrated

to be a reliable tool to model the dynamics of $CO_2$ and exchange between the atmosphere and terrestrial flux. Most importantly, as the online coupled modelling system is able to simulate meteorology and biosphere processes simultaneously, it promotes the opportunity to investigate the interactions between atmospheric mixing and terrestrial flux (Carvalhais et al., 2014;Schimel et al., 2015) while comprehensively considering various factors from both sides that affect $CO_2$ in one coordinate frame, which could be a very helpful tool to support policy makers for balancing short-term carbon cycles at regional scales.


*Data availability*

The modelling output is accessible by contacting the corresponding author (yjjiang@pku.org.cn, xhu@ou.edu)

*Author contributions*

The concept and ideas to design the integrated simulation and observation analysis are devised by YJ, X-MH, and XD. Simulation is performed by X-MH. OCO-2 satellite product is collected and processed by X-MH. CT2019 assimilation data and ground-based observations are collected by XD. Tower measurement is conducted, processed, and analysed by QM, JP, and YJ. Model evaluation is performed by MY. The manuscript is prepared by XD and X-MH with input and feedback from
YJ, MY, QM, JP, and GZ.

*Competing interests*

The authors declare that they have no conflict of interest.

*Acknowledgements*

This work is supported by the Fundamental Research Funds for the Central Universities (14380049) and National Key Research and Development Program of China (2016YFC0201900). We thank NASA and NOAA ESRL for providing the public accessible satellite products and observations used in this study. OCO-2 data was collected through https://co2.jpl.nasa.gov/#mission=OCO-2. We thank Tomohiro Oda for providing Open-Data Inventory for Anthropogenic
Carbon dioxide (ODIAC) emissions. ESRL surface flask $CO_2$ data was downloaded from https://www.esrl.noaa.gov/gmd/dv/data.html. TCCON data was downloaded from https://data.caltech.edu/records/1092. CT2019B results were provided by NOAA ESRL, Boulder, Colorado, USA from the website at http://carbontracker.noaa.govCarbonTracker data was downloaded from https://www.esrl.noaa.gov/gmd/ccgg/carbontracker/download.php.

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

Table 1. Evaluation statistics[1] for WRF-Chem 20km-grid simulation against OCO-2 satellite product at daily intervals

| Season | Vegetation type | Mean Obs. (ppmv) | Mean Sim. (ppmv) | MB[1] (ppmv) | cc[1] | # of samples |
|---|---|---|---|---|---|---|
| Spring | other | 406.85 | 407.81 | 0.96[2] | 0.82 | 16123 |
| | evergreen | 407.52 | 407.89 | 0.36 | 0.73 | 1920 |
| | deciduous | 408.15 | 408.430 | **0.27** | 0.82 | 412 |
| | mixed | 407.79 | 408.21 | 0.41 | 0.79 | 4438 |
| | shrubland | 406.97 | 407.54 | 0.56 | 0.74 | 6550 |
| | savanna | 407.59 | 408.55 | 0.96 | 0.81 | 534 |
| | grass | 406.81 | 407.49 | 0.68 | 0.81 | 11170 |
| | crops | 407.50 | 408.29 | 0.79 | 0.82 | 13548 |
| Summer | other | 403.90 | 404.84 | 0.93 | 0.88 | 13445 |
| | evergreen | 402.68 | 402.24 | -0.44 | 0.85 | 1082 |
| | deciduous | 400.39 | 399.39 | -1.01 | 0.82 | 527 |
| | mixed | 402.04 | 401.60 | -0.43 | 0.87 | 4312 |
| | shrubland | 403.92 | 404.41 | 0.48 | 0.85 | 5193 |
| | savanna | 404.62 | 404.60 | **-0.02** | 0.79 | 170 |
| | grass | 402.35 | 402.66 | 0.31 | 0.88 | 12588 |
| | crops | 402.86 | 403.52 | 0.66 | 0.87 | 7947 |
| Fall | other | 403.32 | 404.35 | 1.03 | 0.82 | 17054 |
| | evergreen | 403.93 | 403.19 | -0.74 | 0.71 | 1716 |
| | deciduous | 403.35 | 403.64 | **0.28** | 0.84 | 281 |
| | mixed | 403.64 | 403.95 | 0.31 | 0.83 | 3611 |
| | shrubland | 403.12 | 404.22 | 1.10 | 0.77 | 8532 |
| | savanna | 403.45 | 404.15 | 0.70 | 0.70 | 504 |
| | grass | 403.22 | 403.65 | 0.43 | 0.85 | 11176 |
| | crops | 403.76 | 404.80 | 1.04 | 0.80 | 13136 |
| Winter | other | 404.76 | 405.80 | 1.03 | 0.80 | 13838 |
| | evergreen | 404.79 | 404.75 | **-0.05** | 0.78 | 2671 |
| | deciduous | 405.38 | 406.65 | 1.27 | 0.79 | 135 |
| | mixed | 405.20 | 405.79 | 0.59 | 0.79 | 2108 |
| | shrubland | 404.76 | 405.84 | 1.09 | 0.79 | 7683 |
| | savanna | 404.63 | 405.83 | 1.20 | 0.75 | 1064 |
| | grass | 405.06 | 405.64 | 0.58 | 0.77 | 5967 |
| | crops | 405.17 | 406.36 | 1.19 | 0.79 | 15508 |

[1] Mean bias was calculated as: $MB = \frac{1}{N}\sum_{i=1}^{N}(Sim_i - Obs_i)$, and correlation coefficient was calculated as: $cc = $

$\frac{\sum_{i=1}^{N}(Sim_i - \bar{Sim})(Obs_i - \bar{Obs})}{\sqrt{\sum_{i=1}^{N}(Sim_i - \bar{Sim})^2}\sqrt{\sum_{i=1}^{N}(Obs_i - \bar{Obs})^2}}$, where $\bar{Sim}$ is the average of simulations, $\bar{Obs}$ is the average of observations.

[2] For each season, evaluation statistic with the worst performance (largest absolute value of MB) is highlighted in red, and the one with best performance (smallest absolute value of MB) is highlighted with in bold font.

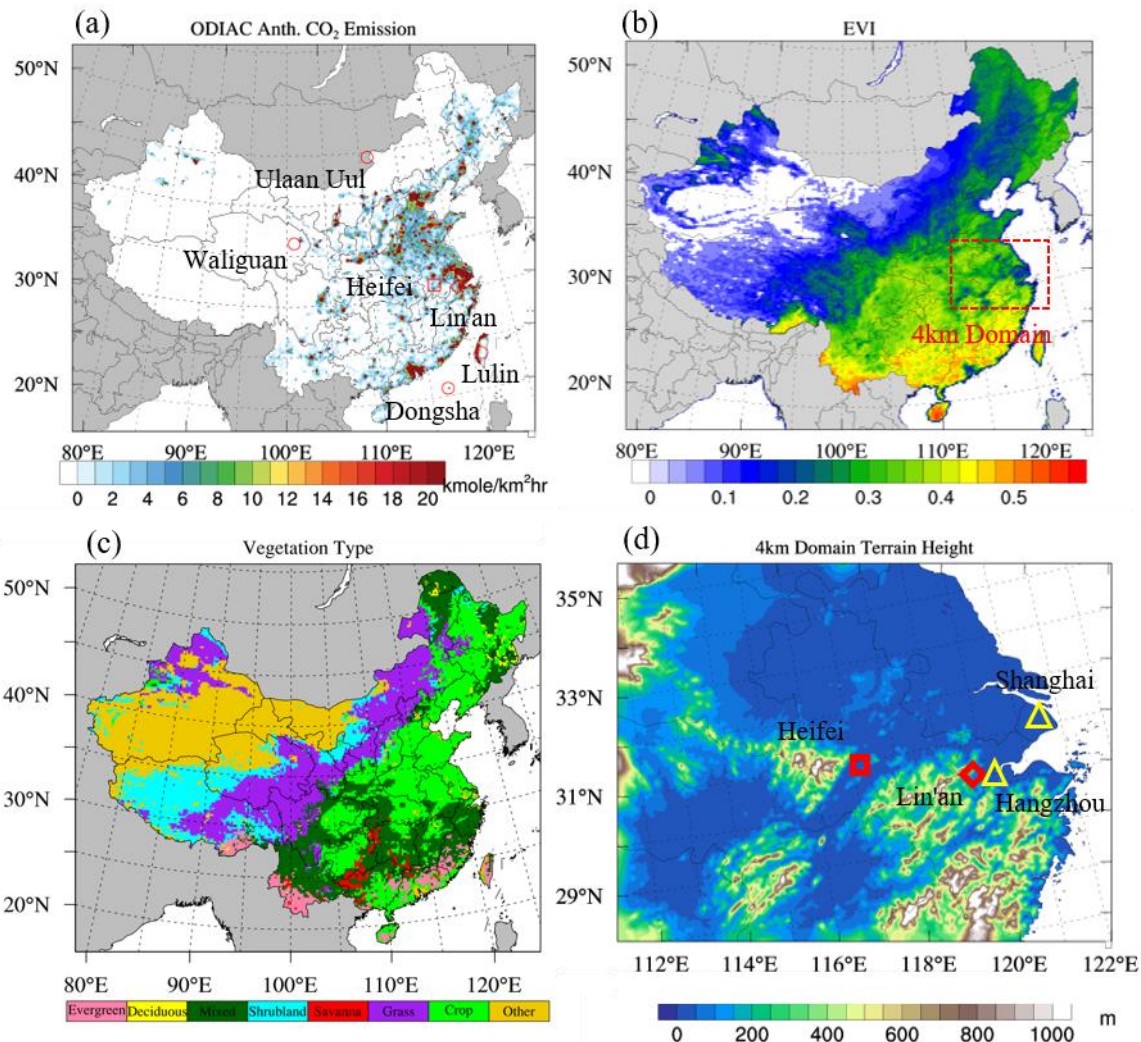

**Figure 1: Annual averages of (a) ODIAC emission, (b) MODIS EVI, and (c) dominant vegetation type** in the **20km simulation domain, and (d) terrain height of the 4km simulation domain. The locations of the ESRL sites, TCCON Hefei site, and Lin'an tower site are indicated with red circles, rectangles, and diamonds respectively in (a). The 4km domain is indicated with the red dash rectangle in (b), and the locations of Hangzhou and Shanghai are indicated with yellow triangles in (d).**


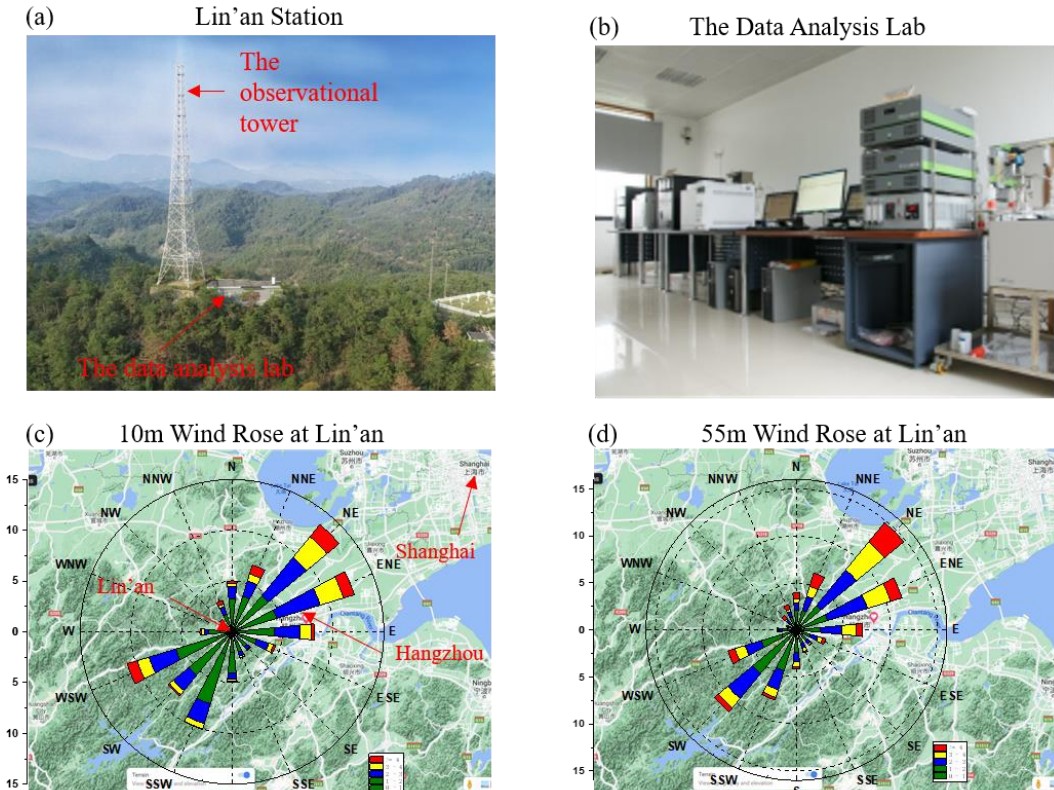

**Figure 2: Photos of the (a) Lin'an regional atmospheric background station and (b) the data analysis lab; and wind rose map at Lin'an derived from wind speed and wind direction observations for 2016-2018 at (c) 10m and (d) 50m.**


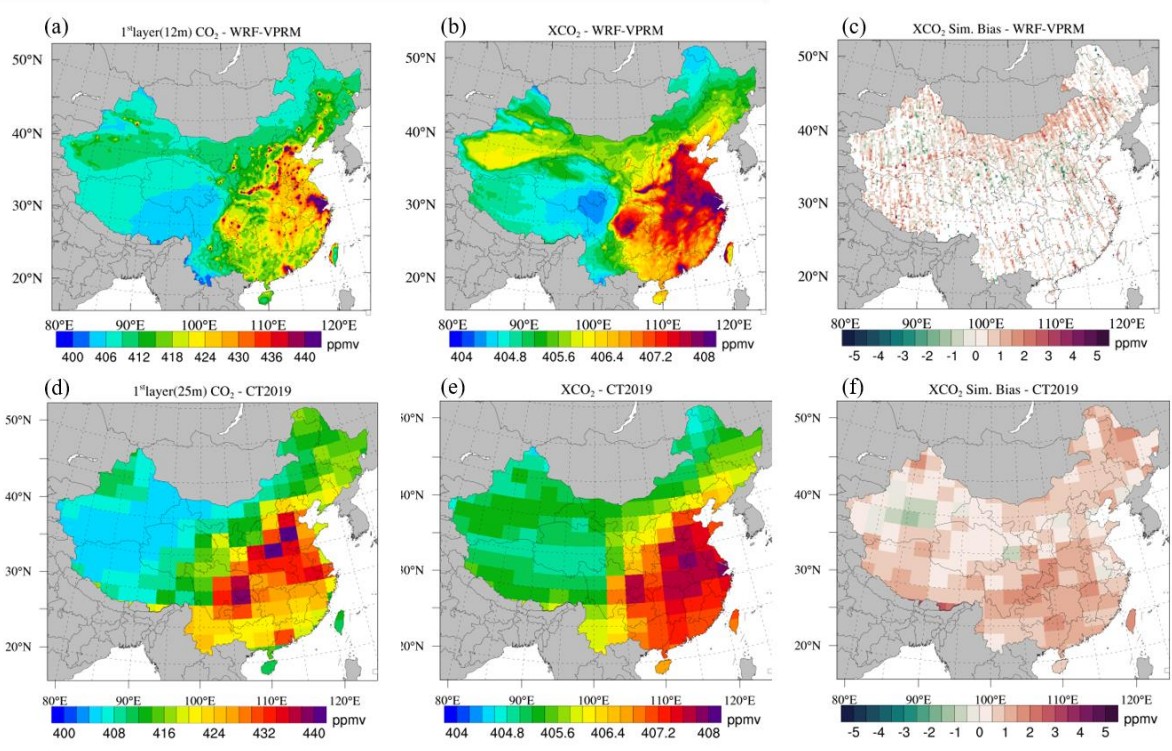

**Figure 3: 2016-2018 averages of WRF-Chem simulations of (a) 1st layer (mid-layer height is 12km) CO₂ concentration, and (b) XCO₂ concentration; (c) WRF-Chem simulated XCO₂ bias against OCO-2; (d)-(f) is same as (a)-(c) but for CT2019 (1st layer mid-level height is 25m).**


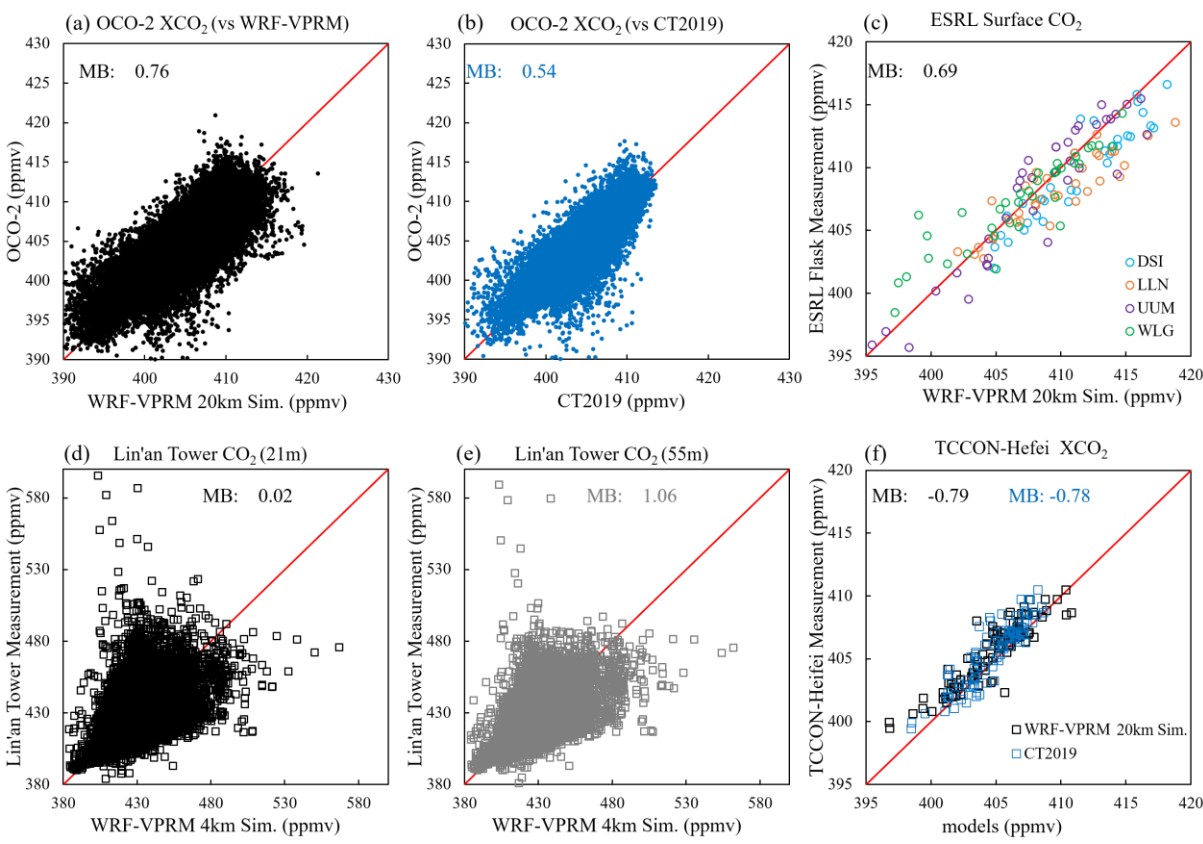

**Figure 4: Data pairs for OCO-2 against (a) WRF-Chem and (b) CT2019; (c) ESRL against WRF-Chem; Lin'an tower against WRF-Chem 4km-grid simulation at (d) 21m and (e) 55m; and (f) TCCON-Hefei against WRF-Chem and CT2019.**

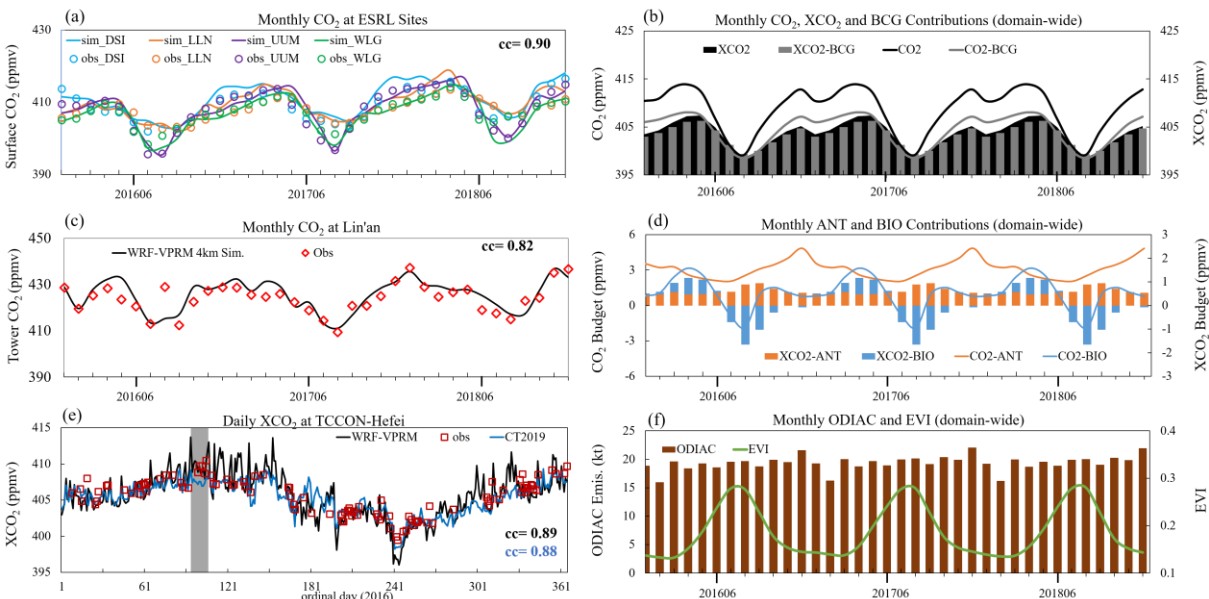

Figure 5: Monthly variations of (a) $CO_2$ at ESRL sites, (b) total (black) and background (BCG, grey) $CO_2$ (line) and $XCO_2$ (area and bar), (c) $CO_2$ at Lin'an station (averaged for daytime 21m and 55m data); (d) contributions from anthropogenic (ANT, orange) and biogenic (BIO, blue) for $CO_2$ (lines) and $XCO_2$ (bars); (f) ODIAC emission and MODIS EVI; and (e) Daily variation of $XCO_2$ at TCCON-Hefei site.

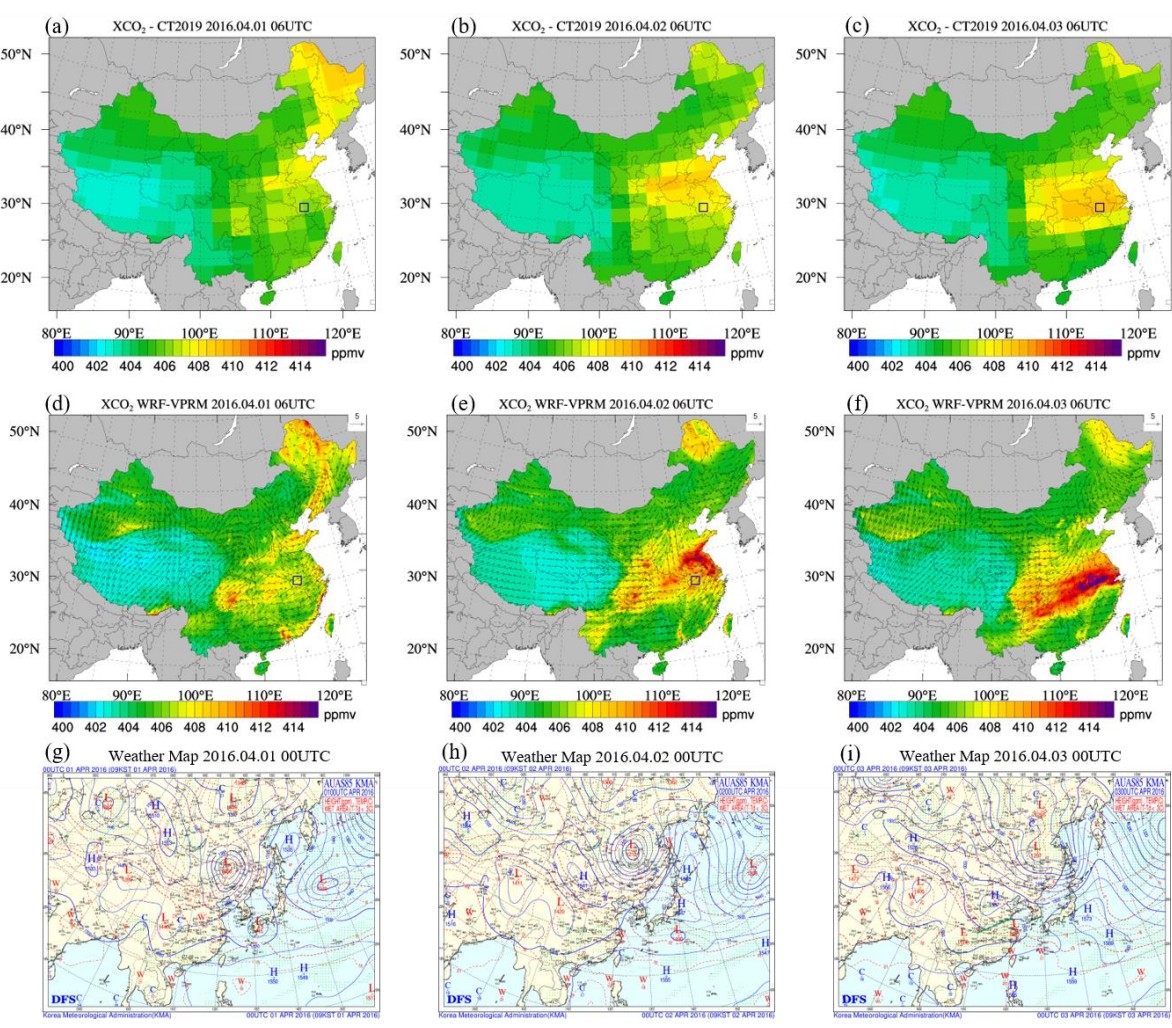

**Figure 6: Daily XCO₂ from CT2019 (a-c) and WRF-Chem (d-f), weather map from Korea Meteorological Administration (g-i). The blue box represents location of Hefei.**

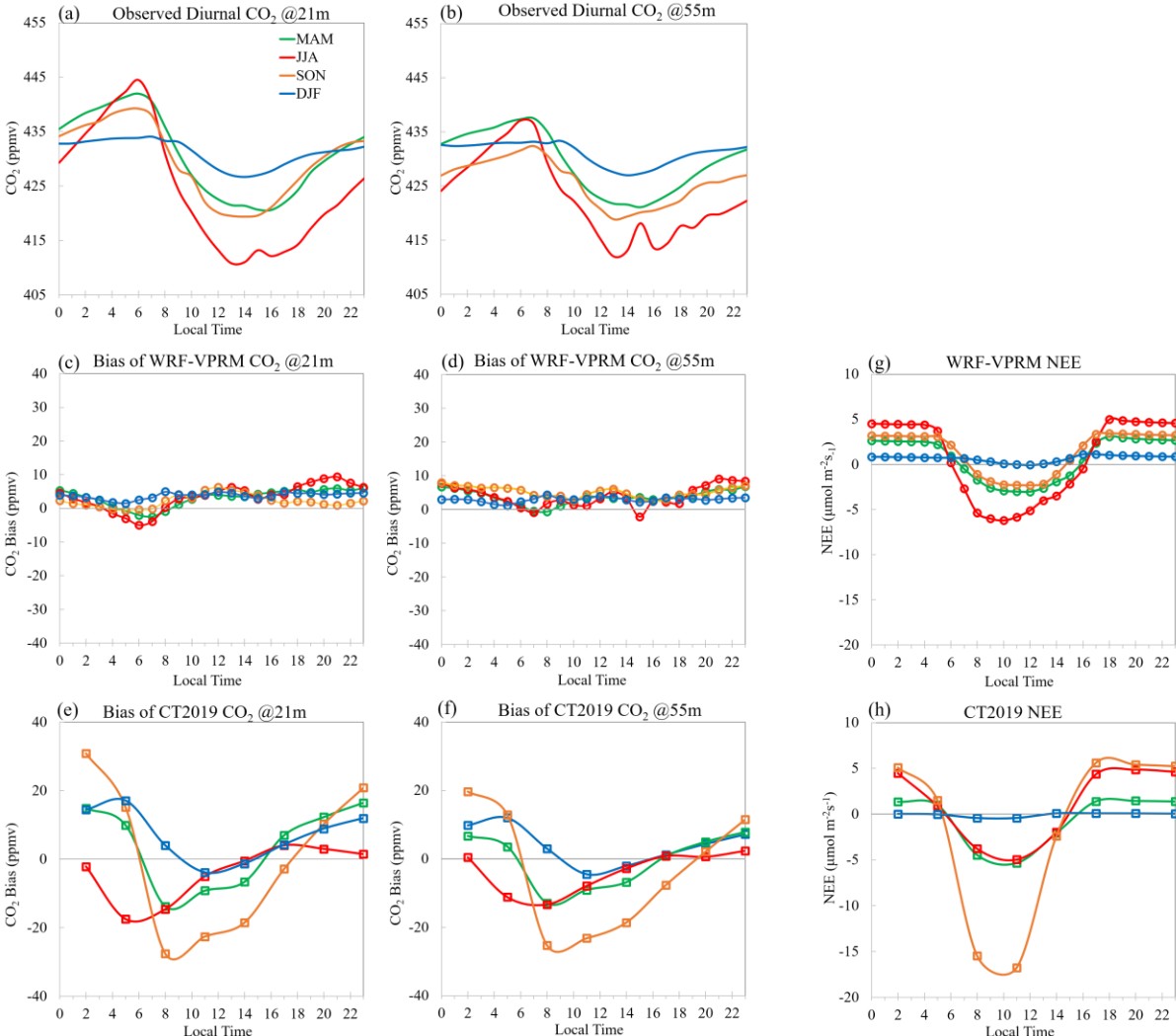

**Figure 7: Seasonal mean diurnal variations of observed CO₂ at (a) 21m and (b) 55m; WRF-Chem simulation biases of CO₂ at (c) 21m and (d) 55m; CT2019 simulated biases at (e) 21m and (f) 55m; Simulated NEE from (g) WRF-Chem and (h) CT2019.**

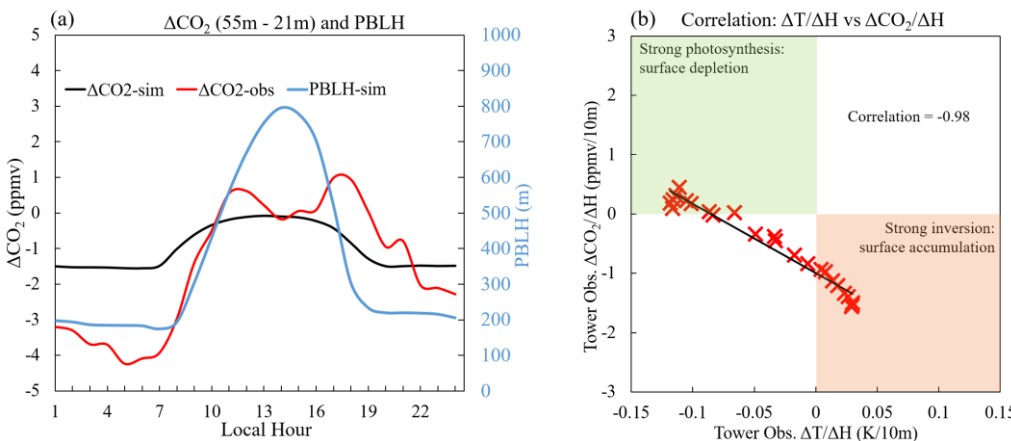

**Figure 8: (a) Average (2016-2018) diurnal variations of simulated (black line) and observed (red line) $\Delta CO_2$ and simulated (blue line) PBLH at Lin'an station; and (b) correlation between $CO_2$ gradient between 55m and 21m ($\Delta CO_2/\Delta H$) and temperature gradient ($\Delta T/\Delta H$) at Lin'an station.**