# Peer review of "Analysis of CO2 spatiotemporal variations in China using a weatherbiosphere-online-coupled model"

_Atmospheric Chemistry and Physics, 2020_

## Referee Comment (RC1) · Anonymous Referee #1 · 4 Jan 2021

In this manuscript, "Analysis of CO2 spatiotemporal variation in China using tower data and a weather-biosphere-online-coupled model, WRF-VPRM", authors analyzed the spatiotemporal variations of CO2 concentrations and fluxes measured at a flux tower (Lin'an), column-concentrations of satellite and flask measurements, and compared with WRF-VPRM simulation results for three years from 2016 to 2018. The study finds that anthropogenic emissions determined the spatial distribution of CO2 and that the observations and simulations showed an increasing trend in XCO2. It also displayed the WRF-VPRM model successfully reproduced both ground and satellite-column CO2 concentrations but was relatively largely overestimated by the flux measurements. The authors have conducted tremendous work on both analyses of various measurement

data and simulations for this study. However, this paper is not proper for publication on Atmospheric Chemistry and Physics, considering the arrangement and completeness of the logical development, and a bit unreasonable application of WRF-VPRM. The detailed comments can be described below.

General comments Authors never reported model configurations, specifically for meteorology, although one of the key works for this study is a numerical simulation. For the publication of modeling work, the basic model configuration for both $CO_2$ and meteorology is essential for the scientific community's numerical experiments' reproducibility. The model configuration is also significant for understanding the analysis of $CO_2$ concentrations near the ground, interacting with the PBL and stable boundary layer (Section 3.3). Without prior knowledge about the model set-up, readers cannot be sure of the quality assurance of simulation results. For the analysis of $CO_2$ concentration near the ground, especially in nighttime, the PBL scheme's choice is significant. As the "first" WRF-VPRM simulation over the study domain, authors must conduct the PBL scheme sensitivity tests and find the best physics scheme combinations before progressing this manuscript. The fine resolution (20-km) is too coarse to capture Lin'an's footprint area, which would be roughly < 4 km at the height level under stable conditions. Therefore, it is hard and a bit unreasonable to directly compare with local-scale measured fluxes. This manuscript's key sites or regions are Lin'an and Hangzhou, but their locations and site descriptions are missing. No mark on maps or description sub-section. This is very important for readers' understanding. In Line 295, for example, the authors tried to describe the transport of $CO_2$ plume from Hangzhou. However, readers do not know their spatial location, so they cannot catch up the further discussion. How far are the two locations? How much is Hangzhou close to efficiently affect to Lin'an? The authors explained in Line 240 that the Lin'an site could be affected by regional anthropogenic emissions. However, readers would not understand which regions or directions could be the main culprit. Therefore, wind direction analysis should be needed in Figure 6, where only wind speeds are displayed. Besides the location of Lin'an, its LULC features should be described in a sub-section. A native English speaker should edit this

[Figure]

paper, especially for tense. Usually, past tense is supposed to be used in the method and the results and discussion sections, especially for the action and experiment have done already.

Specific comments

1) The main title is not proper for summarizing the whole content. Specifically, the first part of the sentence (before 'and') indicated only tower data, although the authors used integrated various measurement data. In the later part, after 'and', the sentence sounds like the WRF-VPRM model analysis, which is odd because we do not usually analyze the model itself. 2) Figures are a bit chaotically mixed, so readers cannot smoothly follow the writing flow. Please explain figure by figure in the body for the consistency of the flow of paragraphs. In Figure 1, for example, the spatial distribution (upper panel) and the photo of the Lin'an site (bottom panel) should be drawn on two different figures. In Figure 6, some sub-figures should also be separated.

Line 92: Add the version of WRF. Line 283: What is $\Delta H$ ? Line 300: Footprints at each level of the flux site should be quantified

Technical corrections Figure 1: Figure 1(f) is missing, although Line 131 referred to it. Figure 4: The graphic resolution is poor for (e). Readers cannot identify or separate the difference between the shaded area and others. Figure 6. The scale of the y-axis must be matched for a clear comparison. Line 162: The full name of NMB is mentioned later, Line 166.

---

## Referee Comment (RC2) · Anonymous Referee #2 · 25 Jan 2021

The paper presents simulations of atmospheric CO2 over China at high spatial resolution and comparisons against observations. Temporal and spatial variations are assessed. The manuscript needs to be revised before it can be recommended for publication.

General Comments

It is stated that WRF has been evaluated extensively with respect to meteorology, but no references are given. In this context an evaluation of the model against meteorological observations within the domain of interest is needed. If no references can be found, this evaluation should be included in this manuscript.

[Figure]

The authors claim that the WRF-VPRM model can be used to assess carbon budgets related to biospheric fluxes and to anthropogenic emissions. However, it should be clear that VPRM is a highly simplified light use efficiency model that represents upscaling of flux observations from eddy covariance measurements made over Europe, which would need further optimization through inverse modelling (see e.g. Kountouris et al., 2018) even for the European domain. Applying the same VPRM parameters to a different domain will result in even larger errors in fluxes. Furthermore, anthropogenic fluxes are simply used as input to WRF-VPRM, thus regional carbon budgets can directly be derived using the emission inventory data themselves.

I fully agree with Reviewer #1 in that more details are needed with respect to the description of the model setup, but also the observation sites. For example, only at the end of the discussion it is mentioned that the city of Hangzhou is located 60 km away from the Lin'an tower. This clearly belongs to the description of the data used, ideally in a specific section within the methods section, entitled for example "atmospheric observations".

Specific comments

In addition to the mean bias (MB), the normalized mean bias (NMB) does not really provide additional information, as the mean of atmospheric $CO_2$ for specific locations/periods is always within 10% of 400 ppm. I therefore suggest not reporting the normalized mean bias.

Abstract: Line 18: "characterize CO2 dynamics" I suggest rephrasing "characterize the dymanics of CO2 in the atmosphere"

Line 23: "determined" rephrase, e.g. "dominated"

Line 44: "calibrated" -> "adjusted"

Line 45: "determine posterior flux"

Line 76: "suffer from" -> "due to"

Line 94: A reference for CT2019 is needed. From where and when were the data downloaded? See also NOAA's usage policy under https://www.esrl.noaa.gov/gmd/ccgg/carbontracker/citation.php

Line 114: "pure" I suggest using "process based"

Line 130 "samplings of CO2 surface concentrations with monthly intervals are collected through" -> "atmospheric samples near the surface are collected at monthly intervals and analysed for CO2 through"

Line 134: please provide a clear reference for the OCO-2 data. From where and when were the data downloaded?

Line 137: please provide a clear reference for the TCCON data from the Hefei site. From where and when were the data downloaded? Please ensure also that the TCCON data use policy is followed (see https://tccon-wiki.caltech.edu/Main/DataUsePolicy).

Line 166: "forest which" -> "forest, which"

Line 187: The attribution of model-observation discrepancy to the vertical allocation of emissions is not plausible. It would be required to at least state the distance of upstream (strong) emission sources that could have an impact on the CO2 profile over the site.

Line 188: "Biosphere models" please rephrase, e.g. "tracer transport models"; also: CASA is a biospheric process model providing biosphere-atmosphere exchange fluxes, to which level within a tracer transport model those are added is not prescribed by CASA. Also note that the injection height is relevant only for anthropogenic emissions of CO2 due to the associated smoke stack height or plume rise (see Brunner et al., 2019), not for biospheric fluxes.

Line 198: "Pu et al. (Pu et al., 2014)" -> "Pu et al. (2014)"

Line 204: Please reformulate, this sentence is not clear. What do you mean by "as a

process-based model"?

Line 209, Fig 4c: I suggest using daytime values at the Lin'an tower. Note that the air samples at the NOAA stations are also taken during daytime, usually in a well-mixed boundary layer. Otherwise nocturnal peaks in (modeled or observed) CO2 will simply dominate.

Line 210:"we will probe into bias" -> "we will discuss details on the bias"

Line 224: "minimums" -> "minima", "maximums" -> "maxima"

Lines 241 – 245: I strongly recommend using ppm/yr as unit for the different trends.

Line 265: "may have also estimated" I assume that there is corresponding output from WRF-VPRM with hourly biosphere fluxes from respiration and photosynthesis, such that it can be confirmed that WRF-VPRM simulates non-zero respiration during non-growing season.

Line 280: "above or underestimation" -> "above, or due to underestimation"

Line 300: it should be made clear that here the concentration footprint is meant, rather than the flux footprint. See e.g. Lin et al. (2003) for concentration footprint, and Schmid et al. (1994) for flux footprint.

References: Kountouris, P., Gerbig, C., Rödenbeck, C., Karstens, U., Koch, T.F., Heimann, M., 2018. Atmospheric $CO_2$ inversions on the mesoscale using data-driven prior uncertainties: quantification of the European terrestrial $CO_2$ fluxes. Atmos. Chem. Phys. 18, 3047–3064. doi:10.5194/acp-18-3047-2018

Lin, J.C., Gerbig, C., Wofsy, S.C., Andrews, A.E., Daube, B.C., Davis, K.J., Grainger, C.A., 2003. A near-field tool for simulating the upstream influence of atmospheric observations: The Stochastic Time-Inverted Lagrangian Transport (STILT) model. J Geophys Res-Atmos 108. doi:10.1029/2002JD003161

Schmid, H.P., 1994. Source Areas for Scalars and Scalar Fluxes. Boundary-Layer Meteorol 67, 293–318. doi:10.1007/BF00713146

---

## Author Response (AR1)

This document summarized point-to-point responses to the comments from two anonymous referees, and a list of main changes in the manuscript.

**Comments and responses to referee#1**

**General comments**: Authors never reported model configurations, specifically for meteorology, although one of the key works for this study is a numerical simulation. For the publication of modeling work, the basic model configuration for both CO2 and meteorology is essential for the scientific community's numerical experiments' reproducibility. The model configuration is also significant for understanding the analysis of CO2 concentrations near the ground, interacting with the PBL and stable boundary layer (Section 3.3). Without prior knowledge about the model set-up, readers cannot be sure of the quality assurance of simulation results. For the analysis of CO2 concentration near the ground, especially in nighttime, the PBL scheme's choice is significant. As the "first" WRF-VPRM simulation over the study domain, authors must conduct the PBL scheme sensitivity tests and find the best physics scheme combinations before progressing this manuscript. The fine resolution (20-km) is too coarse to capture Lin'an's footprint area, which would be roughly < 4 km at the height level under stable conditions. Therefore, it is hard and a bit unreasonable to directly compare with local-scale measured fluxes. This manuscript's key sites or regions are Lin'an and Hangzhou, but their locations and site descriptions are missing. No mark on maps or description sub-section. This is very important for readers' understanding. In Line 295, for example, the authors tried to describe the transport of CO2 plume from Hangzhou. However, readers do not know their spatial location, so they cannot catch up the further discussion. How far are the two locations? How much is Hangzhou close to efficiently affect to Lin'an? The authors explained in Line 240 that the Lin'an site could be affected by regional anthropogenic emissions. However, readers would not understand which regions or directions could be the main culprit. Therefore, wind direction analysis should be needed in Figure 6, where only wind speeds are displayed. Besides the location of Lin'an, its LULC features should be described in a sub-section. A native English speaker should edit this paper, especially for tense. Usually, past tense is supposed to be used in the method and the results and discussion sections, especially for the action and experiment have done already.

**General Response**: We greatly appreciate the referee for his/her time and efforts devoted to the review of our submission. The major comment was the manuscript didn't provide details about the model configuration and tower measurement. The reviewer also questioned the arrangement and completeness of the manuscript, for instance, readers do not know the locations of the Lin'an tower station and Hangzhou thus they cannot catch up the further discussion. We realize that most of the comments are due to the missing of necessary details regarding the modeling method, the observational method, and the study domain. We will present these details in this document as shown in the following responses. The reviewer suggested that the 20km grid resolution simulation may not be directly compared with local-scale flux measurement, thus we conduct a new set of simulation at 4km grid resolution over a smaller domain covering the tower measurement site, and revise the manuscript accordingly. In addition, some of the sub-figures in the original submission have been rearranged and drawn separately, we will show these figures as well.

**Specific comments and responses:**

**Comment#1:** Authors never reported model configurations, specifically for meteorology, although one of the key works for this study is a numerical simulation. For the publication of modeling work, the basic model configuration for both $CO_2$ and meteorology is essential for the scientific community's numerical experiments' reproducibility

**Response**: The model configuration in this study mostly follow the work of Hu et al. (2020), except that Hu et al. (2020) simulated North America but our simulation is over East Asia. Hu et al. (2020) is frequently cited in our manuscript but we forget to mention about the configuration. We apologize for this careless mistake. As a coupled weather-biosphere model, the WRF-VPRM simulation contained two parts of configuration for WRF and VPRM respectively. The configuration on the WRF side is presented in the following table (Table S1 in revised manuscript).

**Table: WRF-VPRM Model Configuration**

| Attribute | Configuration | Reference |
| --- | --- | --- |
| Short wave radiation | Duhia algorithm | Dudhia (1989) |
| Long wave radiation | Rapid radiative transfer model (RRTM) | Mlawer et al. (1997) |
| Boundary layer | Yonsei University (YSU) scheme | - |
| Microphysics | Morrison scheme | Morrison et al. (2009) |
| Cumulus | Grell-3 scheme | Grell and Devenyi (2002) |
| Land surface model | Noah land-surface scheme | Chen and Dudhia (2001) |
| Vertical levels | 47 | - |
| Horizontal resolution | 20 km × 20 km with 234 (south-north) × 285 (west-east) grid points; 4km × 4km with 215 (south-north) × 280 (west-east) grid points | - |
| Time step | 60s | - |
| Meteorological initial and lateral boundary conditions | NCEP/DOE Reanalysis 2 (R2) | - |
| Interior nudging | Spectral nudging | - |
| Nudging variables | horizontal wind components, temperature, and geopotential height | - |
| Nudging coefficient | $3\times10^{-5}\,s^{-1}$ | - |
| Nudging height | above PBL | - |
| Wave number | 5 and 3 in the zonal and meridional directions, respectively | - |

The configuration on the VPRM side refers to emission inputs, initial and boundary conditions, and the parameterization (for $PAR_0, \alpha, \beta, \lambda$). We have described emission inputs and initial and boundary conditions in the manuscript at line#91-97. Physical parameterization followed the default configuration as mentioned at line#112. The values of the default parameterization are presented in the following table (Table S2 in revised manuscript).

**Table: VPRM Parameter Values Used in This Study**

| | evergreen forest | Deciduous forest | Mixed forest | Shrub | Savanna | Crop | Grass |
|---|---|---|---|---|---|---|---|
| $PAR_0$ (µmol PAR·m$^{-2}$·s$^{-1}$) | 745.306 | 514.13 | 419.5 | 590.7 | 600 | 1074.9 | 717.1 |
| $\lambda$ (µmol CO$_2$·m$^{-2}$s$^{-1}$/µmol PAR·m$^{-2}$·s$^{-1}$) | 0.13 | 0.1 | 0.1 | 0.18 | 0.18 | 0.085 | 0.115 |
| $\alpha$ (µmol CO$_2$·m$^{-2}$·s$^{-1}$·°C$^{-1}$) | 0.1247 | 0.092 | 0.2 | 0.0634 | 0.2 | 0.13 | 0.0515 |
| $\beta$ (µmol CO$_2$·m$^{-2}$·s$^{-1}$) | 0.2496 | 0.843 | 0.27248 | 0.2684 | 0.3376 | 0.542 | -0.0986 |

The above tables are included in the "supplement information" of the revised manuscript. We also add necessary description in the main text of the revised manuscript.

We conduct a new set of simulation with 4km grid resolution with exactly the same configuration over a smaller domain, as shown in the following figure (Figure 1(b) and (d) in revised manuscript). The new 4km-grid simulation has a domain size as 215 (south-north) × 280 (west-east) grid points over a significantly smaller domain than the 20km-grid simulation domain. The 4km-grid simulation showed very similar result to the 20km-grid simulation. Thus the 20km-grid simulation was used to characterize the spatiotemporal distributions of CO$_2$ over China, and the 4km-grid simulation was only used to compare with tower data collected at Lin'an tower. Detailed comparison will be shown in the response for comment#3. Our major conclusion was not changed, thus we do not attempt to rerun the whole China domain simulation with 4km grid resolution due to limited computational resource.

[Figure]

**Figure**: 4km-grid simulation domain over Yangtze River Delta (YRD).

Reference:

Chen, F., & Dudhia, J. (2001). Coupling an advanced land surface‑hydrology model with the Penn State‑NCAR MM5 modeling system. Part I: Model implementation and sensitivity. Monthly Weather Review, 129(4), 569–585.

Dudhia, J. (1989). Numerical study of convection observed during the Winter Monsoon Experiment using a mesoscale two‑dimensional model. Journal of the Atmospheric Sciences, 46(20), 3077–3107.

Grell, G. A., & Devenyi, D. (2002). A generalized approach to parameterizing convection combining ensemble and data assimilation techniques. Geophysical Research Letters, 29(14), 1693. https://doi.org/10.1029/2002gl015311

Mlawer, E. J., Taubman, S. J., Brown, P. D., Iacono, M. J., & Clough, S. A. (1997). Radiative transfer for inhomogeneous atmospheres: RRTM, a validated correlated‑k model for the longwave. Journal of Geophysical Research‑Atmospheres, 102(D14), 16663–16682. https://doi.org/10.1029/97jd00237

Morrison, H., Thompson, G., & Tatarskii, V. (2009). Impact of cloud microphysics on the development of trailing stratiform precipitation in a simulated squall line: Comparison of one‑ and two‑moment schemes. Monthly Weather Review, 137(3), 991–1007.

**Comment#2:** The model configuration is also significant for understanding the analysis of $CO_2$ concentrations near the ground, interacting with the PBL and stable boundary layer (Section 3.3). Without prior knowledge about the model set‑up, readers cannot be sure of the quality assurance of simulation results. For the analysis of $CO_2$ concentration near the ground, especially in nighttime, the PBL scheme's choice is significant. As the "first" WRF‑VPRM simulation over the study domain, authors must conduct the PBL scheme sensitivity tests and find the best physics scheme combinations before progressing this manuscript.

**Response**: We totally agree with the comment that PBL scheme plays a very important role in model simulation. Selection of PBL schemes is critical for accurate simulation of lower tropospheric $CO_2$ vertical distribution as shown in previous studies (Ballav et al., 2016; Diaz‑Isaac et al., 2018). Our study applies the YSU scheme based on a thorough investigation of the YSU scheme (Hu et al., 2010; Hu et al., 2012; Hu et al., 2013; Hu et al., 2019) and a test comparison with the MYJ scheme. The YSU scheme has been shown to perform well for both daytime and nighttime at the 20 km grid spacings used in this study (Hu et al., 2012; Yang et al., 2019). The YSU scheme is a nonlocal scheme with explicit treatment of entrainment fluxes, which was shown to be critical to reproducing convective boundary layer structures (Hu et al., 2013) and achieve a better performance than some local schemes such as the Mellor–Yamada–Janjic (MYJ) scheme (Wang et al., 2016). For stable boundary layer, an update in stability function in 2013 led to a better YSU performance in terms of reproducing nighttime profiles of both meteorological and chemical variables, particularly over the Great Plains (Wang et al., 2016). YSU led to a better $CO_2$ simulation than MYJ in our earlier WRF‑VPRM application over the U.S. domain (Hu et al., 2020), thus it is chosen in this study. The YSU scheme has been demonstrated to perform well as one of the best options over East Asia for both air quality modeling (Huang et al., 2014; Liu et al., 2016; Wang et al., 2020) and meteorology modeling studies (Cheng et al., 2014; Tao et al., 2011).

Reference:

Ballav, S., Patra, P. K., Sawa, Y., Matsueda, H., Adachi, A., Onogi, S., De, U. K. (2016). Simulation of CO2concentrations at Tsukuba tall tower using WRF-CO2tracer transport model. Journal of Earth System Science, 125(1), 47-64. 10.1007/s12040-015-0653-y

Diaz-Isaac, L. I., Lauvaux, T., & Davis, K. J. (2018). Impact of physical parameterizations and initial conditions on simulated atmospheric transport and CO2 mole fractions in the US Midwest. Atmospheric Chemistry and Physics, 18(20), 14813-14835. 10.5194/acp-18-14813-2018

Hu, X.-M., Doughty, D. C., Sanchez, K. J., Joseph, E., & Fuentes, J. D. (2012). Ozone variability in the atmospheric boundary layer in Maryland and its implications for vertical transport model. Atmospheric Environment, 46, 354-364. DOI 10.1016/j.atmosenv.2011.09.054

Hu, X.-M., Klein, P. M., & Xue, M. (2013). Evaluation of the updated YSU planetary boundary layer scheme within WRF for wind resource and air quality assessments. Journal of Geophysical Research-Atmospheres, 118(18), 10490-10505. 10.1002/jgrd.50823

Hu, X.-M., Nielsen-Gammon, J. W., & Zhang, F. Q. (2010). Evaluation of Three Planetary Boundary Layer Schemes in the WRF Model. Journal of Applied Meteorology and Climatology, 49(9), 1831-1844. 10.1175/2010jamc2432.1

Hu, X.-M., Xue, M., & Li, X. (2019). The Use of High-Resolution Sounding Data to Evaluate and Optimize Nonlocal PBL Schemes for Simulating the Slightly Stable Upper Convective Boundary Layer. Monthly Weather Review, 147(10), 3825-3841. 10.1175/mwr-d-19-0085.1

Hu, X. M., Crowell, S., Wang, Q., Zhang, Y., Davis, K. J., Xue, M., . . . DiGangi, J. P. (2020). Dynamical Downscaling of CO2 in 2016 Over the Contiguous United States Using WRF-VPRM, a Weather-Biosphere-Online-Coupled Model. Journal of Advances in Modeling Earth Systems, 12(4), e2019MS001875. 10.1029/2019ms001875

Wang, W. G., Shen, X. Y., & Huang, W. Y. (2016). A Comparison of Boundary-Layer Characteristics Simulated Using Different Parametrization Schemes. Boundary-Layer Meteorology, 161(2), 375-403. 10.1007/s10546-016-0175-4

Yang, Y., Hu, X.-M., Gao, S., & Wang, Y. (2019). Sensitivity of WRF simulations with the YSU PBL scheme to the lowest model level height for a sea fog event over the Yellow Sea. Atmospheric Research, 215, 253-267. https://doi.org/10.1016/j.atmosres.2018.09.004

**Comment#3:** The fine resolution (20-km) is too coarse to capture Lin'an's footprint area, which would be roughly < 4 km at the height level under stable conditions. Therefore, it is hard and a bit unreasonable to directly compare with local-scale measured fluxes.

**Response**: We agree with the reviewer that grid resolution matters for air quality and meteorology simulations. We conduct a new set of simulation with 4km grid resolution over a smaller domain covering the TCCON-Hefei site and Lin'an tower site, and compare it with the 20km-grid simulation. In general, the 4km-grid simulation showed well consistent result with the 20km grid simulation over the same area, and no different conclusion could be drawn from the new set of simulation. The following figure (Figure S1 in revised manuscript) presents the spatial distributions of $CO_2$ and $XCO_2$ from the two sets of simulations. The 4km-grid

simulation provided a more detailed presentation of the spatial distribution, but the levels of $CO_2$ and $XCO_2$ were quite close to the 20km-grid resolution, thus most of the discussions within the revised manuscript still use 20km-grid simulation data, only the discussion related with Lin'an tower data used the new 4km-grid simulation data.

[Figure]

**Figure**: Annual averaged $CO_2$ (left column) and $XCO_2$ (right column) from WRF-VPRM 4km-grid simulation (top row) and 20km-grid simulation (bottom row). Locations of Hefei and Lin'an are presented with red rectangle and diamond.

For $XCO_2$ simulations, we find that the two sets of simulations differed by only 0.1 ppmv (<0.03%) at the TCCON-Hefei site. The following figure (Figure S2(a) in revised manuscript) presents the comparison of daily $XCO_2$ between 20km-grid and 4-km grid simulations at TCCON-Hefei site. The two simulations showed fairly close results.

[Figure]

**Figure:** Comparison of 20km-grid and 4km-grid simulations at TCCON Hefei site.

For $CO_2$ simulations however, the 4km-grid simulation showed much smaller bias than the 20km-grid simulation at Lin'an tower for $CO_2$, thus we update within the manuscript to use 4km-grid simulation to compare with the Lin'an observation, as shown in the following figure (Figure 4(d) and (e) in revised manuscript).

[Figure]

**Figure**: WRF-VPRM 4km-grid simulation evaluated against Lin'an tower observations at 21m (left) and 55m (right).

**Comment#4:** This manuscript's key sites or regions are Lin'an and Hangzhou, but their locations and site descriptions are missing. No mark on maps or description sub-section. This is very important for readers' understanding. In Line 295, for example, the authors tried to describe the transport of $CO_2$ plume from Hangzhou. However, readers do not know their spatial location, so they cannot catch up the further discussion. How far are the two locations? How much is Hangzhou close to efficiently affect to Lin'an? The authors explained in Line 240 that the Lin'an site could be affected by regional anthropogenic emissions. However, readers would not understand which regions or directions could be the main culprit. Therefore, wind direction analysis should be needed in Figure 6, where only wind speeds are displayed. Besides the location of Lin'an, its LULC features should be described in a sub-section.

**Response**: We agree with the referee that more details are necessary to demonstrate the locations of Lin'an and Hangzhou, especially for those unfamiliar with China. Lin'an is a district of Hangzhou city. The Lin'an Regional Atmospheric Background Station is about 60km west to the downtown center of Hangzhou. To show these details, we add the description of the location in the revised manuscript at line#129-130. We also include the following figure (Figure 2(c) in revised manuscript) to demonstrate the locations of Lin'an and downtown center of Hangzhou, and also demonstrate the prevailing wind at Lin'an. To demonstrate the wind speed as well as the wind direction, the wind rose map was derived from hourly observations of 10m and 55m wind speed and wind direction at Lin'an for 2016-2018. It shows the prevailing wind directions at Lin'an are northeast and southwest.

[Figure]

[Figure]

**Figure**: Wind rose map derived from Lin'an tower hourly observations of 10m (left) and 55m (right) wind speed and wind directions for 2016-2018.

**Comment#5:** A native English speaker should edit this paper, especially for tense. Usually, past tense is supposed to be used in the method and the results and discussion sections, especially for the action and experiment have done already.

**Response**: As recommended by the editor during our initial submission, the manuscript has been carefully edited by a native speaker, and a lot of grammar typos have been corrected before the open discussion. We have changed to past tense for the descriptions of modeling method. Most of the discussions have also been changed to past tense. Full version of revised manuscript is not allowed to be submitted during the open discussion, thus we list some some examples as below:

At line#88: "Both simulations were configured with 47 vertical layers with model tops at 10hPa."
At line#126: "Hourly measurements of CO2 concentrations were collected at the Lin'an Regional Atmospheric Background Station …"
At line#188: "Evaluation at the Lin'an station was performed with the 4km-grid simulation"
At line#251: "WRF-VPRM reproduced the trends in good agreement with ground and satellite observations."
At line#269: "We find that both models prominently overestimated during nighttime, which shall be attributed to the bias in simulating NEE"

**Comment#6:** The main title is not proper for summarizing the whole content. Specifically, the first part of the sentence (before 'and') indicated only tower data, although the authors used integrated various measurement data. In the later part, after 'and', the sentence sounds like the WRF-VPRM model analysis, which is odd because we do not usually analyze the model itself.

**Response**: We agree with the referee that the original title emphasized too much on the tower measurement. We have revised the title as: "Analysis of $CO_2$ spatiotemporal variations in China using a weather-biosphere-online-coupled model"

**Comment#7:** Figures are a bit chaotically mixed, so readers cannot smoothly follow the writing flow. Please explain figure by figure in the body for the consistency of the flow of paragraphs. In Figure 1, for example, the spatial distribution (upper panel) and the photo of the Lin'an site (bottom panel) should be drawn on two different figures. In Figure 6, some

sub-figures should also be separated.

**Response**: We agree with the referee that some of the figures contain too many sub-figures which may not belong to the same category. According to this comment, we have separated Figure 1 into two different figures (Figure 1 and Figure 2 in revised manuscript) to show the simulation domains and photos of Lin'an station separately. We also rearranged Figure 6 (Figure 7 in revised manuscript) as recommended by the reviewer as shown in response for comment#13.

**Comment#8:** Line 92: Add the version of WRF.

**Response**: We use WRF Version 3.9.1.1 for the WRF-VPRM model simulation. We have included this information at line#86 in the revised manuscript.

**Comment#9:** Line 283: What is ΔH?

**Response**: ΔH stands for the above ground height difference between the two levels being investigated. For the Lin'an $CO_2$ concentration observations, ΔH stands for the height difference between the 55m and 21m monitors, thus ΔH is 34m. We appreciate the referee for point out this issue, and we have included this information at line#292 in the revised manuscript as: "Fig.8(b) presents the correlation between air temperature gradient (ΔT/ΔH) and $CO_2$ concentration gradient (Δ$CO_2$/ΔH) calculated with annual averaged diurnal tower observations, where ΔT, Δ$CO_2$, and ΔH represents the difference of air temperature, $CO_2$ concentration, and height between the two tower levels. The temperature gradient and $CO_2$ concentration gradient clearly demonstrate the influence of boundary layer stability on the $CO_2$ vertical profile." We would also like to mention that in the original version of Fig.7(b) (now is Fig.8(b) in the revised manuscript), we used annual averaged diurnal data for each year to calculate the gradients, thus there were 24 data points for each year and there were 72 data points in the figure. But we just realize that it was not consistent with Fig.7(a) which showed diurnal profiles averaged for all three years. So, we calculate the gradients from diurnal data averaged for all three years thus there are only 24 data points in revised manuscript, and the correlation is calculated as -0.98.

**Comment#10:** Line 300: Footprints at each level of the flux site should be quantified

**Response**: We agree with the referee that showing the footprints would be a straightforward demonstration to support the discussion regarding transport impact, but unfortunately there was no wind speed and wind direction measurement at 21m of the Lin'an tower. We only have wind observations at 10m and 55m. We apply the method proposed by Hsieh et al. (2000) to calculate footprints at these two levels. The scalar flux $(F)$ and the footprint $(f)$ are related by (equation 1 in Hsieh et al. (2000)):

$$F(x, z_m) = \int_{-\infty}^{x} S(x) f(x, z_m)\, dx$$

where $S\ (unit: g\ m^{-2}\ s^{-1})$ is the source strength, $z_m$ is the measurement height, and the mean wind direction is along the horizontal coordinate, $x$. Based on this method, we calculated the peak location of the footprint ($x_{f=f_{peak}}$, equation 19 in Hsieh et al. (2000)) and the location where the fetch-to-height ratio equals 90% ($x_{F/S_0=0.9}$, equation 20 in Hsieh et al. (2000)) at 10m and 55m respectively as shown in the following figure. We applied the CALMET

model (Scire et al., 1998) to calculate related variables such as friction velocity and sensible heat.

[Figure]

**Figure**: Locations where footprint reaches peak value (left); Locations where the fetch-to-height ratio equals 90%. Both units are meters.

The above figure demonstrates that upper air (55m) received influences from prominently longer distances than lower air (10m). Considering the dominant upwind directions are northwest at Lin'an tower (figure in response to comment#4), it's likely that 55m at Lin'an had larger footprints than 21m from Hangzhou. Footprint was mentioned at line#296 and line#300 in the original manuscript. In that paragraph, we attempted to demonstrate that the boundary layer stability was closely correlated with the $CO_2$ concentration gradient. Footprint was mentioned to further the discussion by demonstrating that 55m received more influence from Hangzhou than 21m. We assumed that upper air usually has larger footprint than lower air. However, this comment reminds us that we are not able to solidly demonstrate it because no wind speed and wind direction measurement was available at 21m. Thus we decide to remove the discussion regarding footprint (line#295-302 in original manuscript) in this revision, and our main conclusion in this paragraph remains unchanged.

Reference:
Hsieh, C.I., Katul, G., Chi, T.W: An approximate analytical model for footprint estimation of scalar fluxes in thermally stratified atmospheric flows, Advances in Water Resources, 23, 765-772, 2000.
Scire, J.S., Robe, F.R., Fernau, M.E., Yamartino R.J.: A user's Guide for the CALMET Meteorological Model (Version 5) Earth Tech Inc, Concord, MA (1998)

**Comment#11:** Figure 1: Figure 1(f) is missing, although Line 131 referred to it.
**Response**: We apologize for this careless typo. We have split the original Figure 1 into two figures as suggested by comment#7. We have revised the description as: "Flask samplings of $CO_2$ surface with monthly intervals are collected through the National Oceanic and Atmospheric Administration's (NOAA's) Earth System Research Laboratory (ESRL) at four sites (shown in Fig. 1(a)) within our modeling domain."

**Comment#12:** Figure 4: The graphic resolution is poor for (e). Readers cannot identify or separate the difference between the shaded area and others.

**Response**: We apologize for this careless mistake. The original figure has been automatically compressed in the .docx document. We have turned off the "automatic compress" option in Microsoft-Word software, and updated all figures with high resolution in the revised manuscript.

**Comment#13** Figure 6. The scale of the y-axis must be matched for a clear comparison.

**Response**: The y-axis in Figure 6 has been adjusted accordingly as shown in the following figure (Figure 7 in revised manuscript). As recommended in comment#7, we reorganize the figure by removing the wind speed figure (c) as shown below. We also use wider distance between the $CO_2$ concentration figures (a-f) and NEE figures (g and h).

[Figure]

**Figure** 7: Seasonal mean diurnal variations of observed $CO_2$ at (a) 21m and (b) 55m; WRF-VPRM simulation biases of $CO_2$ at (c) 21m and (d) 55m; CT2019 simulated biases at (e) 21m and (f) 55m; Simulated NEE from (g) WRF-VPRM and (h) CT2019.

As the wind directions were already shown, we add a new figure (Figure S3 in the revised manuscript) to demonstrate the comparison of wind speed between 10m and 55m at Lin'an tower, as shown below.

[Figure]

**Figure**: Observed diurnal profiles of wind speed at Lin'an.

**Comment#14** Line 162: The full name of NMB is mentioned later, Line 166.
**Response**: We appreciate the reviewer for pointing this out. In the original submission we actually mentioned the full name twice at line#161 and line#166 respectively, we will remove the full name "normalized mean bias" at line#166 in the revised manuscript.

**Comments and responses to referee#2**

**General comments and responses:**

**General Comment#1**: It is stated that WRF has been evaluated extensively with respect to meteorology, but no references are given. In this context an evaluation of the model against meteorological observations within the domain of interest is needed. If no references can be found, this evaluation should be included in this manuscript.

**Response**: We agree with the referee that necessary references should be provided regarding the performance of WRF over China. We have added the following references at line#147 in the revised manuscript. These references were selected as representative because they applied the similar versions or configurations of WRF during their simulations. These recent publications provided detailed evaluations and demonstrations of the meteorology simulation performance of WRF in China.

Reference:

Gao, Y. Q., Lee, X. H., Liu, S. D., Hu, N., Hu, C., Liu, C., Zhang, Z., and Yang, Y. C.: Spatiotemporal variability of the near-surface $CO_2$ concentration across an industrial-urban-rural transect, Nanjing, China, Sci Total Environ, 631-632, 1192-1200, 2018.

Tang, J. P., Niu, X. R., Wang, S. Y., Gao, H. X., Wang, X. Y., and Wu, J.: Statistical downscaling and dynamical downscaling of regional climate in China: Present climate evaluations and future climate projections, J Geophys Res-Atmos, 121, 2110-2129, 2016.

Wang, W. G., Shen, X. Y., & Huang, W. Y. (2016). A Comparison of Boundary-Layer Characteristics Simulated Using Different Parametrization Schemes. Boundary-Layer Meteorology, 161(2), 375-403. 10.1007/s10546-016-0175-4

Yang, Y., Hu, X.-M., Gao, S., & Wang, Y. (2019). Sensitivity of WRF simulations with the YSU PBL scheme to the lowest model level height for a sea fog event over the Yellow Sea. Atmospheric Research, 215, 253-267. https://doi.org/10.1016/j.atmosres.2018.09.004

**General Comment#2**: The authors claim that the WRF-VPRM model can be used to assess carbon budgets related to biospheric fluxes and to anthropogenic emissions. However, it should be clear that VPRM is a highly simplified light use efficiency model that represents upscaling of flux observations from eddy covariance measurements made over Europe, which would need further optimization through inverse modelling (see e.g. Kountouris et al., 2018) even for the European domain. Applying the same VPRM parameters to a different domain will result in even larger errors in fluxes. Furthermore, anthropogenic fluxes are simply used as input to WRF-VPRM, thus regional carbon budgets can directly be derived using the emission inventory data themselves.

**Response**: We totally agree with the referee that applying the same VPRM parameters to different domains will result in uncertainties, we have added Kountouris et al. (2018) as a reference to support the discussion of model uncertainty. We mentioned the parameterization issue at several places in the original manuscript (for instance, line#111,

173, 176 ), and pointed it out in the "Conclusion" section (line#337) that VPRM parameterization need further improvement. To our knowledge, WRF-VPRM was first applied to Europe domain by Ahmadov et al. (2007), and the parameters followed Mahadevan et al. (2008) which were derived from eddy covariance measurements collected at 22 towers in North America (5 towers in Canada and 17 towers in United States, see Table1 in Mahadevan et al., 2008). Ahmadov et al. (2007) mentioned that the parameters were slightly modified but the values were not reported. For our study, we used the calibrate VPRM parameters for different vegetation types by using observed NEE from a group of 65 eddy covariance tower sites over North America, and using these parameters over US domain also has uncertainties (Hu et al., 2020). For WRF-VPRM application in China, Li et al. (2020) evaluated the parameterization with eddy covariance data collected at two sites in northeast China, and reported that the default parameterization can successfully reproduce the temporal variations and intensity of biospheric fluxes, but also pointed out that the bias over mixed forest site should be due to the VPRM parameterization. Unfortunately, Lin'an tower didn't have eddy covariance measurements to support the modification of VPRM parameters, so we tried to use the hourly $CO_2$ concentration measurements to reveal the uncertainty of the model. It is true the parameters can be further calibrated using tower flux data over China, and we hope there will be more efforts devoted to collect such data over different land categories in the near future. Inverse modeling is certainly one of the options to help verify or indicate the uncertainty of biospheric model, but it also retains uncertainty such as being sensitive to the formulation of prior flux. Thus research efforts are needed to improve both of them as has been recognized by the community (Kondo et al., 2020). Justification/evaluation of an appropriate prior flux from WRF-VPRM over China is one of the objectives of this study. Following Li et al. (2020) which was the first study discussing VPRM uncertainties associated with parameterization in northeast China, this study intended to investigate the case in south China with Lin'an tower data, which is a necessary step for future inverse calibration or calibration using flux tower data. Despite the uncertainties associated with the VPRM parameters, Li et al. (2020) and this study demonstrated that WRF-VPRM captured many characteristics of $CO_2$ fluxes/concentrations, including seasonal/episodic/diurnal variation of fluxes/concentrations.

We acknowledge the uncertainties associated with WRF-VPRM, that is why we'd like to use tower data to evaluate and understand the uncertainties in this study, which could guide future calibration of VPRM in the region. In terms of anthropogenic flux, the anthropogenic emission int this study was from Open-Data Inventory for Anthropogenic Carbon dioxide (ODIAC) emission version 2018, which had its own uncertainties and cannot be treated as truth. All the WRF-VPRM uncertainties associated with fluxes must be evaluated/examined by more atmospheric observations, this study is just one of such attempts. We fully agree with the referee that it is necessary to improve VPRM parameterization based on local eddy covariance data, and pointing this out is one of the objectives of this study. In fact, East Asia is one of the regions having largest uncertainty in $CO_2$ budget estimation (Kondo et al., 2020). Our study is one of the attempts to help improve the understanding in this area with biospheric modeling method, and certainly more observational and modeling efforts are necessary to reduce the uncertainty in the

future. To address the reviewer's concerns, we emphasized these points in the revised manuscript.

Reference:

Ahmadov, R., Gerbig, C., Kretschmer, R., Koerner, S., Neininger, B., Dolman, A. J., and Sarrat, C.: Mesoscale covariance of transport and CO2 fluxes: Evidence from observations and simulations using the WRF-VPRM coupled atmosphere-biosphere model, J Geophys Res-Atmos, 112, 2007.

Hu, X. M., Crowell, S., Wang, Q. Y., Zhang, Y., Davis, K. J., Xue, M., Xiao, X. M., Moore, B., Wu, X. C., Choi, Y., and DiGangi, J. P.: Dynamical Downscaling of CO2 in 2016 Over the Contiguous United States Using WRF-VPRM, a Weather-Biosphere-Online-Coupled Model, Jounal of Advances in Modeling Earth Systems, 12, 10.1029/2019MS001875, 2020.

Kondo, M., Patra, P. K., Sitch, S., Friedlingstein, P., Poulter, B., Chevallier, F., Ciais, P., Canadell, J. G., Bastos, A., Lauerwald, R., Calle, L., Ichii, K., Anthoni, P., Arneth, A., Haverd, V., Jain, A. K., Kato, E., Kautz, M., Law, R. M., Lienert, S., Lombardozzi, D., Maki, T., Nakamura, T., Peylin, P., Rodenbeck, C., Zhuravlev, R., Saeki, T., Tian, H. Q., Zhu, D., and Ziehn, T.: State of the science in reconciling top-down and bottom-up approaches for terrestrial CO2 budget, Global Change Biol, 26, 1068-1084, 2020.

Li, X. L., Hu, X. M., Cai, C. J., Jia, Q. Y., Zhang, Y., Liu, J. M., Xue, M., Xu, J. M., Wen, R. H., and Crowell, S. M. R.: Terrestrial CO2 Fluxes, Concentrations, Sources and Budget in Northeast China: Observational and Modeling Studies, J Geophys Res-Atmos, 125, 2020.

Mahadevan, P., Wofsy, S. C., Matross, D. M., Xiao, X. M., Dunn, A. L., Lin, J. C., Gerbig, C., Munger, J. W., Chow, V. Y., and Gottlieb, E. W.: A satellite-based biosphere parameterization for net ecosystem CO2 exchange: Vegetation Photosynthesis and Respiration Model (VPRM), Global Biogeochem Cy, 22, 2008

**General Comment#3**: I fully agree with Reviewer #1 in that more details are needed with respect to the description of the model setup, but also the observation sites. For example, only at the end of the discussion it is mentioned that the city of Hangzhou is located 60 km away from the Lin'an tower. This clearly belongs to the description of the data used, ideally in a specific section within the methods section, entitled for example "atmospheric observations".

**Response**: We agree with the two referees that more details are necessary to provide a clear description of the observational sites and modeling method. Regarding the details of Lin'an site, the following figure had been added to show the locations of Lin'an tower, downtown Hangzhou, and Shanghai, along with wind rose figures to demonstrate the prevailing winds at 10m and 55m at Lin'an tower. We have also added a few more sentences in the revised manuscript (line#131-137) describing the location and prevailing winds at Lin'an tower.

[Figure]

**Figure** : Photos of the (a) Lin'an regional atmospheric background station and (b) the data analysis lab; and wind rose map at Lin'an derived from wind speed and wind direction observations for 2016-2018 at (c) 10m and (d) 50m.

Regarding the modelling method, we have added the following table in the revised manuscript to provide a detailed description of the model configuration. Necessary descriptions of the WRF configuration were also added in the main text (line#90-93). Considering that the WRF configuration was popular in China and most WRF users may be quite familiar with it, the table was added into the supplementary material. The configuration of VPRM parameterization was also added into the supplementary material.

**Table S1. WRF-VPRM Model Configuration**

| Attribute | Configuration | Reference |
|---|---|---|
| Short wave radiation | Duhia algorithm | Dudhia (1989) |
| Long wave radiation | Rapid radiative transfer model (RRTM) | Mlawer et al. (1997) |
| Boundary layer | Yonsei University (YSU) scheme | Hong et al. (2006) |
| Microphysics | Morrison scheme | Morrison et al. (2009) |
| Cumulus | Grell-3 scheme | Grell and Devenyi (2002) |
| Land surface model | Noah land-surface scheme | Chen and Dudhia (2001) |
| Vertical levels | 47 | - |
| Model top | 10hPa | |
| Horizontal resolution | 20 km × 20 km with 234 (south-north) × 285 (west-east) grid points; 4km × 4km with 215 | - |

| | | |
|---|---|---|
| | (south-north) × 280 (west-east) grid points | |
| Time step | 60s | - |
| Meteorological initial and lateral boundary conditions | NCEP/DOE Reanalysis 2 (R2) | - |
| Interior nudging | Spectral nudging | - |
| Nudging variables | horizontal wind components, temperature, and geopotential height | - |
| Nudging coefficient | $3\times10^{-5}$ $s^{-1}$ | - |
| Nudging height | above PBL | - |
| Wave number | 5 and 3 in the zonal and meridional directions, respectively | - |

**Table S2. VPRM Parameter Values Used in This Study**

| | Evergreen forest | Deciduous forest | Mixed forest | Shrub | Savanna | Crop | Grass |
|---|---|---|---|---|---|---|---|
| $PAR_0$ ($\mu$mol PAR$\cdot$m$^{-2}\cdot$s$^{-1}$) | 745.306 | 514.13 | 419.5 | 590.7 | 600 | 1074.9 | 717.1 |
| $\lambda$ ($\mu$mol $CO_2\cdot$m$^{-2}$s$^{-1}$/$\mu$mol PAR$\cdot$m$^{-2}\cdot$s$^{-1}$) | 0.13 | 0.1 | 0.1 | 0.18 | 0.18 | 0.085 | 0.115 |
| $\alpha$ ($\mu$mol $CO_2\cdot$m$^{-2}\cdot$s$^{-1}\cdot$°C$^{-1}$) | 0.1247 | 0.092 | 0.2 | 0.0634 | 0.2 | 0.13 | 0.0515 |
| $\beta$ ($\mu$mol $CO_2\cdot$m$^{-2}$s$^{-1}$) | 0.2496 | 0.843 | 0.27248 | 0.2684 | 0.3376 | 0.542 | -0.0986 |

Reference:

Chen, F., and Dudhia, J.: Coupling an advanced land surface-hydrology model with the Penn State-NCAR MM5 modeling system. Part I: Model implementation and sensitivity, Mon Weather Rev, 129, 569-585, 2001.

Dudhia, J.: Numerical Study of Convection Observed during the Winter Monsoon Experiment Using a Mesoscale Two-Dimensional Model, J Atmos Sci, 46, 3077-3107, 1989.

Grell, G. A., and Devenyi, D.: A generalized approach to parameterizing convection combining ensemble and data assimilation techniques, Geophys Res Lett, 29, 2002.

Hong, S. Y., Noh, Y., and Dudhia, J.: A new vertical diffusion package with an explicit treatment of entrainment processes, Mon Weather Rev, 134, 2318-2341, 2006

Mlawer, E. J., Taubman, S. J., Brown, P. D., Iacono, M. J., and Clough, S. A.: Radiative transfer for inhomogeneous atmospheres: RRTM, a validated correlated-k model for the longwave, J Geophys Res-Atmos, 102, 16663-16682, 1997.

Morrison, H., Thompson, G., and Tatarskii, V.: Impact of Cloud Microphysics on the Development of Trailing Stratiform Precipitation in a Simulated Squall Line: Comparison of One- and Two-Moment Schemes, Mon Weather Rev, 137, 991-1007, 2009.

**Specific comments and responses:**

**Comment#1:** In addition to the mean bias (MB), the normalized mean bias (NMB) does not really provide additional information, as the mean of atmospheric CO2 for specific locations/periods is always within 10% of 400 ppm. I therefore suggest not reporting the normalized mean bias.
**Response**: We agree with the referee that normalized mean bias doesn't provide additional information as the atmospheric CO2 concentration is close to 400ppm, thus the relative bias can always be generally estimated with the mean bias. We have removed the values of normalized mean bias in the revised manuscript.

**Comment#2:** Abstract: Line 18: "characterize CO2 dynamics" I suggest rephrasing "characterize the dynamics of CO2 in the atmosphere"
**Response**: We appreciate the referee for the detailed writing suggestions, we have rephrased it as suggested in the revised manuscript.

**Comment#3:** Line 23: "determined" rephrase, e.g. "dominated"
**Response**: It has been rephrased in the revised manuscript.

**Comment#4:** Line 44: "calibrated" -> "adjusted"
**Response**: It has been rephrased in the revised manuscript.

**Comment#5:** Line 45: "determine posterior flux"
**Response**: "terrestrial" has been removed in the revised manuscript as suggested.

**Comment#6:** Line 76: "suffer from" -> "due to"
**Response**: It has been rephrased in the revised manuscript.

**Comment#7:** Line 94: A reference for CT2019 is needed. From where and when were the data downloaded? See also NOAA's usage policy under https://www.esrl.noaa.gov/gmd/ccgg/carbontracker/citation.php
**Response**: We greatly appreciate the referee for this comment. We have added the following text in the acknowledge and Jacobson et al. (2020) by following the NOAA's usage policy.
Text added in the acknowledgement: "CT2019B results were provided by NOAA ESRL, Boulder, Colorado, USA from the website at http://carbontracker.noaa.gov . CarbonTracker data was downloaded from https://www.esrl.noaa.gov/gmd/ccgg/carbontracker/download.php. "
Reference:
Jacobson, A. R., Schuldt, K. N., Miller, J. B., Oda, T., Tans, P., Andrews, A., Mund, J., Ott, L., Collatz,G. J., Aalto, T., Afshar, S., Aikin, K., Aoki, S., Apadula, F., Baier, B., Bergamaschi, P., Beyersdorf, A., Biraud, S. C., Bollenbacher, A., Bowling, D., Brailsford, G., Abshire, J. B., Chen, G., Chen, H., Chmura, L., Colomb, A., Conil, S., Cox, A., Cristofanelli, P., Cuevas, E., Curcoll, R., Sloop, C. D., Davis, K., Wekker, S.

D., Delmotte, M., DiGangi, J. P., Dlugokencky, E., Ehleringer, J., Elkins, J. W., Emmenegger, L., Fischer, M. L., Forster, G., Frumau, A., Galkowski, M., Gatti, L. V., Gloor, E., Griffis, T., Hammer, S., Haszpra, L., Hatakka, J., Heliasz, M., Hensen, A., Hermanssen, O., Hintsa, E., Holst, J., Jaffe, D., Karion, A., Kawa, S. R., Keeling, R., Keronen, P., Kolari, P., Kominkova, K., Kort, E., Krummel, P., Kubistin, D., Labuschagne, C., Langenfelds, R., Laurent, O., Laurila, T., Lauvaux, T., Law, B., Lee, J., Lehner, I., Leuenberger, M., Levin, I., Levula, J., Lin, J., Lindauer, M., Loh, Z., Lopez, M., Luijkx, I. T., Lund Myhre, C., Machida, T., Mammarella, I., Manca, G., Manning, A., Marek, M. V., Marklund, P., Martin, M. Y., Matsueda, H., McKain, K., Meijer, H., Meinhardt, F., Miles, N., Miller, C. E., Molder, M., Montzka, S., Moore, F., Morgui, J.-A., Morimoto, S., Munger, B., Necki, J., Newman, S., Nichol, S., Niwa, Y., ODoherty, S., Ottosson-Lofvenius, M., Paplawsky, B., Peischl, J., Peltola, O., Pichon, J.-M., Piper, S., Plass-Dolmer, C., Ramonet, M., Reyes-Sanchez, E., Richardson, S., Riris, H., Ryerson, T., Saito, K., Sargent, M., Sasakawa, M., Sawa, Y., Say, D., Scheeren, B., Schmidt, M., Schmidt, A., Schumacher, M., Shepson, P., Shook, M., Stanley, K., Steinbacher, M., Stephens, B., Sweeney, C., Thoning, K., Torn, M., Turnbull, J., Tørseth, K., Bulk, P. V. D., Dinther, D. V., Vermeulen, A., Viner, B., Vitkova, G., Walker, S., Weyrauch, D., Wofsy, S., Worthy, D., Young, D., and Zimnoch, M.. : CarbonTracker CT2019B, DOI: 10.25925/20201008, 2020.

**Comment#8:** Line 114: "pure" I suggest using "process based"
**Response**: It has been rephrased in the revised manuscript.

**Comment#9:** Line 130 "samplings of CO2 surface concentrations with monthly intervals are collected through" -> "atmospheric samples near the surface are collected at monthly intervals and analysed for CO2 through"
**Response**: It has been rephrased in the revised manuscript.

**Comment#10:** Line 134: please provide a clear reference for the OCO-2 data. From where and when were the data downloaded?
**Response**: The reference for the OCO-2 data is: Kiel et al., (2019). The download link of the OCO-2 data was provided in the "Acknowledgement" as suggested by the journal guidance. The link was: https://co2.jpl.nasa.gov/#mission=OCO-2 (the download options for NETCDF4 or HDF5 format of the OCO-2 data were provided at the bottom of the webpage)
Reference:
Kiel, M., O'Dell, C. W., Fisher, B., Eldering, A., Nassar, R., MacDonald, C. G., and Wennberg, P. O.: How bias correction goes wrong: measurement of X-CO2 affected by erroneous surface pressure estimates, Atmos Meas Tech, 12, 2241-2259, 2019.

**Comment#11:** Line 137: please provide a clear reference for the TCCON data from the Hefei site. From where and when were the data downloaded? Please ensure also that the TCCON data use policy is followed (see https://tccon-wiki.caltech.edu/Main/DataUsePolicy).

**Response**: We greatly appreciate the referee for this reminder. Description of the TCCON-Hefei site and data was provided in Wang et al. (2017), and this publication was included in our reference. We also briefly describe it in the revised manuscript as:

"Daily ground-based Fourier transform spectrometer (FTS) Measured $XCO_2$ at Hefei site (31.90°N, 117.17°E) was also collected through the Total Carbon Column Observing Network (TCCON) for year 2016 (Wang et al., 2017). The TCCON-Hefei site was located in the northwestern rural area of Hefei city and measurements were conducted from September 2015 to December 2016."

We also add the DOI of TCCON-Hefei data (Liu et a., 2018) as required by the usage policy.

Reference:

Liu, C., Wang, W., Sun, Y.: TCCON data from Hefei, China, Release GGG2014R0. TCCON data archive, hosted by CaltechDATA, California Institute of Technology, Pasadena, CA, U.S.A., http://dx.doi.org/10.14291/tccon.ggg2014.hefei01.R0, 2018.

Wang, W., Tian, Y., Liu, C., Sun, Y. W., Liu, W. Q., Xie, P. H., Liu, J. G., Xu, J., Morino, I., Velazco, V. A., Griffith, D. T., Notholt, J., and Warneke, T.: Investigating the performance of a greenhouse gas observatory in Hefei, China, Atmos Meas Tech, 10, 2627-2643, 2017.

**Comment#12:** Line 166: "forest which" -> "forest, which"
**Response**: The whole sentence was rephrased as: "Regarding vegetation type, the model showed the largest MB over deciduous forest of -1.01 and 1.27 ppmv in summer and winter, respectively, which only covered a very small portion in northeast China."

**Comment#13:** Line 187: The attribution of model-observation discrepancy to the vertical allocation of emissions is not plausible. It would be required to at least state the distance of upstream (strong) emission sources that could have an impact on the CO2 profile over the site.
**Response**: We agree with the referee that it is not plausible to saying the vertical allocation of emission is responsible for the model-observation discrepancy without detailed discussion. Other factors such as the parameterization of VPRM and the anthropogenic emission intensity may also contribute to the discrepancy. We have rephrased the statement as "The discrepancy is likely due to the combined effect of vertical allocation of anthropogenic emission and parameterization of VPRM". WRF-VPRM showed prominent better agreement with observations at the ESRL sites in remote areas than Lin'an tower. The major differences between ESRL sites and Lin'an are the vegetation types and geolocations. Validation against the OCO-2 data suggested that WRF-VPRM didn't show significantly different performance over different vegetation types, thus we have rephrased the discussion as anthropogenic emission allocation may play an important role because Lin'an was close to downtown centers while the ESRL sites were located in real remote regions far from anthropogenic emissions as shown in the following figure (Figure 1(a) in the revised manuscript).

[Figure]

**Figure**: Anthropogenic emissions of CO2 and the locations of ground measurement sites

The referee pointed out a very interesting question to state the distance of upstream (strong) emission sources that could have an impact on the CO2 profile. Apparently the ESRL sites were all far from anthropogenic emission sources at local scale so we haven't probe into this issue. For instance, Lulin site (LLN) was located in the Lulin Mountain in central Taiwan with 2826 sea level height, while the anthropogenic emission sources in urban areas were mostly along the west coast, thus the regional anthropogenic emission can hardly affect CO2 profile at Lulin site. Based on the observations available, we checked the footprints (as recommended by the other referee) at Lin'an tower to identify the contributions from different distances as shown in the following figure. Footprints were calculated following the method proposed by Hsieh et al. (2000).

[Figure]

**Figure**: Locations where footprint reaches peak value (left); Locations where the fetch-to-height ratio equals 90%. Both units are meters.

The above figure shows the peak locations (left) of footprint and the location where the fetch-to-height ratio equals 90% (right) at 10m (blue lines) and 55m (orange lines) of Lin'an tower respectively. At 55m height, the peak location of footprints were about 1.2km from NNE, NE, ENE, and E directions. The location of fetch-to-height ratio equals to 90%

were about 22km, suggesting that the upwind areas within this distance contribute 90% to the 55m height at Lin'an tower. This footprint can serve as one example to indicate the distance that upwind sources may affect the $CO_2$ profile. More details of the footprint calculation and discussion were presented in the response for comment#10 for the other referee (https://editor.copernicus.org/index.php/acp-2020-1128-AC1.pdf?_mdl=msover_md&_jrl=10&_lcm=oc108lcm109w&_acm=get_comm_file&_ms=90658&c=196331&salt=840354016545253938). As we were comparing $CO_2$ at 21m and 55m but there was no wind data at 21m, we decided not to include the discussion of footprint in the manuscript but only provide it here.

**Comment#14:** Line 188: "Biosphere models" please rephrase, e.g. "tracer transport models"; also: CASA is a biospheric process model providing biosphere-atmosphere exchange fluxes, to which level within a tracer transport model those are added is not prescribed by CASA. Also note that the injection height is relevant only for anthropogenic emissions of $CO_2$ due to the associated smoke stack height or plume rise (see Brunner et al., 2019), not for biospheric fluxes.
**Response**: The sentence has been rephrased as suggested in the manuscript. We appreciate the referee's detailed comment and discussion of CASA. We also agree that emission injection height is relevant only for anthropogenic emission, and we thank the referee for reminding us to rephrase the writing to avoid misunderstanding.

**Comment#15:** Line 198: "Pu et al. (Pu et al., 2014)" -> "Pu et al. (2014)"
**Response**: The citation has been reformatted in the manuscript.

**Comment#16:** Line 204: Please reformulate, this sentence is not clear. What do you mean by "as a process-based model"?
**Response**: In the original manuscript, we intended to emphasize that WRF-VPRM can simulate atmospheric $CO_2$ without a prior flux input. We realize that the sentence is redundant, and have removed it from the manuscript.

**Comment#17:** Line 209, Fig 4c: I suggest using daytime values at the Lin'an tower. Note that the air samples at the NOAA stations are also taken during daytime, usually in a well-mixed boundary layer. Otherwise nocturnal peaks in (modeled or observed) $CO_2$ will simply dominate.
**Response**: We greatly appreciate the referee for this helpful comment. We have updated the figure and related discussion with daytime data from observation and model. The correlation was increased from 0.77 to 0.82. Fig 4c in the original manuscript was Fig.5(c) in the revised manuscript, as shown below.

[Figure]

**Figure :** Monthly variations of (a) CO2 at ESRL sites, (b) total (black) and background (BCG, grey) CO2 (line) and XCO2 (area and bar), (c) CO2 at Lin'an station (averaged for daytime 21m and 55m data); (d) contributions from anthropogenic (ANT, orange) and biogenic (BIO, blue) for CO2 (lines) and XCO2 (bars); (f) ODIAC emission and MODIS EVI; and (e) Daily variation of XCO2 at TCCON-Hefei site.

**Comment#18:** Line 210:"we will probe into bias" -> "we will discuss details on the bias"
**Response**: We greatly appreciate the referee for the detailed writing suggestions. Discussion of the simulation bias at Lin'an has been revised based on a new set of 4km grid resolution simulation over a smaller domain covering Lin'an. And the original sentence at line#210 has been removed.

**Comment#19:** Line 224: "minimums" -> "minima", "maximums" -> "maxima"
**Response**: These words have been replaced as suggested in the full manuscript.

**Comment#20:** Lines 241 – 245: I strongly recommend using ppm/yr as unit for the different trends.
**Response**: We have used ppmv/yr as the main unit for the different rends in the revised manuscript when it is possible. We kept the percentage unit for some of the descriptions about model simulated XCO2 budgets trends (original line#243-245) because the ppmv values were too small for anthropogenic and biogenic contributions. For instance, the annual average contribution of XCO2-ANT to the budget was 0.59ppmv, thus the trend of XCO2-ANT was 0.0047 ppmv/yr (0.81%/yr), and it may not appropriate to use ppbv for describing CO2, so we kept the usage of percentage for this description.

**Comment#21:** Line 265: "may have also estimated" I assume that there is corresponding output from WRF-VPRM with hourly biosphere fluxes from respiration and photosynthesis, such that it can be confirmed that WRF-VPRM simulates non-zero respiration during nongrowing season.

**Response**: Yes, WRF-VPRM did provide hourly outputs of respiration and photosynthesis uptake. In this sentence (line#265) we intended to say "may overestimate the nighttime respiration". We apologize for the typo induced misunderstanding. We had no flux measurements at Lin'an thus unfortunately we cannot evaluate if the nighttime respiration was overestimated. Li et al. (2020) validated the hourly respiration with eddy covariance data at a mixed forest site Wuying (47.15°N, 131.94°E), so we compared the simulated respiration and photosynthesis uptake between Wuying and Lin'an to indicate that the model may overestimate respiration in warmer areas where VPRM did calculate respiration as non-zero during nongrowing season, as shown in the following figure (Figure S5 in the supplementary material).

[Figure]

**Figure**: Comparison of WRF-VPRM simulated daily variations of biospheric fluxes (left column) and meteorology (right column) between Wuying (top row) and Lin'an (bottom row).

**Comment#22:** Line 280: "above or underestimation" -> "above, or due to underestimation"
**Response**: It has been rephrased in the revised manuscript, we greatly appreciate the referee for the detailed writing suggestions.

**Comment#23:** Line 300: it should be made clear that here the concentration footprint is meant, rather than the flux footprint. See e.g. Lin et al. (2003) for concentration footprint,

and Schmid et al. (1994) for flux footprint.

**Response**: We appreciate the help from the referee to pointing out the difference between concentration footprint and flux footprint. The discussion of footprint was removed from the manuscript mainly because there was no wind data at 21m height.

**List of main changes in the revised manuscript:**

Line#1: The title of the manuscript was changed according to referee#1's specific comment#6.

Line#87-93: The method section was updated by adding the model configuration of WRF. A new set of simulation at 4km grid resolution was also added, and related discussions in sections 3.1 and 3.2 were updated.

Line#580: The original Figure 1 was reorganized as Figure 1 and Figure 2 in the revised manuscript. The locations of the observation stations are marked, the locations of Lin'an tower, downtown Hangzhou, and Shanghai were presented in a regional map, and wind rose map were shown to demonstrate the prevailing winds at Lin'an tower.

Line#610: The original Figure 6 was reorganized to keep the scales consistent between sub-figures.

We also add a new file for supplementary information along with this revision.

---

## Author Response (AR2)

**Point to Point Reply**

**Referee#1 :** The authors have done a great job in revising the manuscript. They have appropriately addressed the issues I raised in the review. I have no further suggestions, and recommend accepting as is.

**Response**: We greatly appreciate the referee for this time and efforts devoted to help us improve the manuscript.

**Referee#2:** The paper is focused on high-resolution simulations of CO2 mixing ratios over China during 2016-2018. A coupled atmospheric tracer model WRF-Chem was used to conduct CO2 simulations. I have a few comments to improve the quality of the paper.

**Comment#1**: The VPRM parameterization is included as one of the chemistry/tracer options within the WRF-Chem model. Therefore it's more accurate to call the model "WRF-Chem" rather than "WRF-VPRM".

**Response**: We appreciate the reviewer for pointing this detail out, we have revised to WRF-Chem in the manuscript. We mentioned the model in the abstract as: "In this study we apply the WRF-Chem model configured with the Vegetation Photosynthesis and Respiration Model (VPRM) option for biomass fluxes in China to characterize the dynamics of $CO_2$ in the atmosphere."

**Comment#2**: Lines 50-55: One of the objectives of the previous WRF-Chem (VPRM) modeling studies was to improve the simulations of mesoscale transport of atmospheric $CO_2$ by simulating meteorology, $CO_2$ fluxes, and transport in a tightly coupled model and high-resolution domain. The importance of capturing the mesoscale $CO_2$ transport in regional/local scales and significant improvements, which were demonstrated by previous WRF-Chem modeling studies (cited here) are not emphasized here.

**Response**: We agree with the referee that WRF-Chem has been demonstrated to successfully capture the mesoscale $CO_2$ transport through previous studies. We have added a few more modeling studies (Beck et al., 2013; Park et al., 2020; Pillai et al., 2012) into lines 50-55 and added the following emphasis into the manuscript:

"Previous modelling studies (Ahmadov et al., 2009;Kretschmer et al., 2012;Park et al., 2018;Beck et al., 2013;Park et al., 2020;Pillai et al., 2012) have demonstrated the weather-biosphere coupled model can successfully capture the mesoscale $CO_2$ transport at regional and local scales with significant improvements."

*Additional References*:

Beck, V., Gerbig, C., Koch, T., Bela, M. M., Longo, K. M., Freitas, S. R., Kaplan, J. O., Prigent, C., Bergamaschi, P., and Heimann, M.: WRF-Chem simulations in the Amazon region during wet and dry season transitions: evaluation of methane models and wetland inundation maps, Atmos Chem Phys, 13, 7961-7982, 2013.

Park, C., Park, S. Y., Gurney, K. R., Gerbig, C., DiGangi, J. P., Choi, Y., and Lee, H. W.: Numerical simulation of atmospheric CO2 concentration and flux over the Korean Peninsula using WRF-VPRM model during Korus-AQ 2016 campaign, Plos One, 15, 2020.

Pillai, D., Gerbig, C., Kretschmer, R., Beck, V., Karstens, U., Neininger, B., and Heimann, M.: Comparing Lagrangian and Eulerian models for CO2 transport - a step towards Bayesian inverse modeling using WRF/STILT-VPRM, Atmos Chem Phys, 12, 8979-8991, 2012.

**Comment#3**: Anthropogenic $CO_2$ emissions: Does the ODIAC emission inventory include hourly, day to day and seasonal variabilities? How the uncertainties in temporal variability of the anthropogenic $CO_2$ emissions affect the conclusions of the study?

**Response**: ODIAC provided monthly emissions, and the hourly scaling factor was

recommended to follow the TIMES (Temporal Improvements for Modeling Emissions by Scaling) developed by Nassar et al. (2013) from the Vulcan emission product (Gurney et al., 2009) and EDGAR (The Emission Database for Global Atmospheric Research: http://edgar.jrc.ec.europa.eu/). Uncertainty in temporal variability of anthropogenic emission may induce bigger bias over urban area than rural are, as we also noticed that the simulation bias was larger at Lin'an than the ESRL sites (Figure 4). But our discussion (Line#300-321 and Figure 8) demonstrated that the simulation can reproduce diurnal pattern of $CO_2$ gradient at Lin'an, suggesting that the uncertainty in temporal variability of anthropogenic emission shall be unimportant. In-depth analysis and quantification of the associated uncertainty would require a modeling study with bottom-up local inventory which is currently unavailable. Considering the relatively smaller contribution and less variability of anthropogenic emission than the biosphere flux (Figure 5(d) and Figure 7), the uncertainty within temporal variability of anthropogenic emission shall not change the conclusion of this study.

**Comment#4**: The ODIAC $CO_2$ emissions are mostly based on the space-based nighttime light data. Thus, it may be less accurate compared to other fuel/energy-based emissions inventories. This needs to be discussed in the paper.

**Response**: ODIAC was developed through an integration of the CDIAC (Carbon Dioxide Information Analysis Center) global and national fossil fuel emission estimates, BP (British Petroleum) statistical review of world energy, power plants geolocation information from the CARMA (Carbon Monitoring and Action) global power plant database, and the satellite observed nightlight data as the referee mentioned. Essentially the fundamental fossil fuel emission estimate was from CDIAC, which was based on the United Nation Energy Statistics Database (Boden et al., 2017). The satellite nighttime light data was used mainly for constrain and spatial disaggregation. ODIAC was compared with another popular inventory EDGAR (The Emission Database for Global Atmospheric Research: http://edgar.jrc.ec.europa.eu/), and they show very close agreement (Figure 2 in Oda et al., 2018). Modeling study actually suggested better performance with ODIAC than EDGAR over the United States (Hu et al., 2020) According to this comment, we have included a brief discussion in the revised manuscript as: "ODIAC has been widely applied in recent modelling studies and demonstrated good agreement with other global inventories (Hedelius et al., 2017;Hu et al., 2020)."

*Reference:*

Oda, T., Maksyutov, S., and Andres, R. J.: The Open-source Data Inventory for Anthropogenic CO2, version 2016 (ODIAC2016): a global monthly fossil fuel CO2 gridded emissions data product for tracer transport simulations and surface flux inversions, Earth Syst Sci Data, 10, 87-107, 2018.

Hedelius, J. K., Feng, S., Roehl, C. M., Wunch, D., Hillyard, P., Podolske, J. R., Iraci, L. T., Patarasuk, R., Rao, P., O'Keeffe, D., Gurney, K. R., Lauvaux, T., and Wennberg, P. O.: Emissions and topographic effects on column CO2 (X-CO2) variations, with a focus on the Southern California Megacity, J Geophys Res-Atmos, 122, 7200-7215, 2017.

Hu, X. M., Crowell, S., Wang, Q. Y., Zhang, Y., Davis, K. J., Xue, M., Xiao, X. M., Moore, B., Wu, X. C., Choi, Y., and DiGangi, J. P.: Dynamical Downscaling of CO2 in 2016 Over the Contiguous United

States Using WRF-VPRM, a Weather-Biosphere-Online-Coupled Model, Jounal of Advances in Modeling Earth Systems, 12, 10.1029/2019MS001875, 2020.

**Comment#5**: 330: Fix "WRV"
**Response**: We thank the referee for pointing out this typo, it has been fixed in the revised manuscript.